# RobustLight: Improving Robustness via Diffusion Reinforcement Learning for Traffic Signal Control

Mingyuan Li [1]   Jiahao Wang [1]   Guangsheng Yu [2]   Xu Wang [2]   Qianrun Chen [3]   Wei Ni [2]   Lixiang Li [1]
Haipeng Peng [1]

## Abstract

Reinforcement Learning (RL) optimizes Traffic Signal Control (TSC) to reduce congestion and emissions, but real-world TSC systems face challenges like adversarial attacks and missing data, leading to incorrect signal decisions and increased congestion. Existing methods, limited to offline data predictions, address only one issue and fail to meet TSC's dynamic, real-time needs. We propose RobustLight, a novel framework with an enhanced, plug-and-play diffusion model to improve TSC robustness against noise, missing data, and complex patterns by restoring attacked data. RobustLight integrates two algorithms to recover original data states without altering existing TSC platforms. Using a dynamic state infilling algorithm, it trains the diffusion model online. Experiments on real-world datasets show RobustLight improves recovery performance by up to 50.43% compared to baseline scenarios. It effectively counters diverse adversarial attacks and missing data. The relevant datasets and code are available at GitHub.

## 1. Introduction

### 1.1. Motivation

Improving traffic efficiency through Traffic Signal Control (TSC) has been established as an effective strategy (Wei et al., 2019b). Traditional TSC systems depend heavily on static, predefined expert system controls, lacking the flexibility of dynamically responding to fluctuating traffic conditions (Lowrie, 1990; Hunt et al., 1982; Webster, 1958). Recently, by integrating the methods of Reinforcement Learning (RL), TSC has demonstrated its superiority

to conventional expert systems in improving vehicular traffic flow (Wei et al., 2018). Deploying TSC algorithms in real-world settings typically involves using sensors, such as cameras and radar, to monitor traffic states, including vehicle counts and speeds. Sensors exposed in public areas are susceptible to noise attacked (Chen et al., 2018; Chowdhury et al., 2023b), leading to potential adversarial attacks. Moreover, in extremely adverse weather conditions, these sensors are prone to physical damage (Laszka et al., 2016b; Lee & Park, 2012). This study delves into these sensor security challenges in real-world TSC systems.

Adversarial attacks on the sensors can range from Gaussian noise and uniformly distributed random noise (U-rand) to more sophisticated strategies, including maximum action-difference attack (MAD) and minimum Q-value attack (MinQ) (Tang et al., 2016). When testing a TSC algorithm, its performance can significantly decline (Gershenson, 2004; Chen et al., 2020; Zhang et al., 2022b) within the CityFlow (Zhang et al., 2019) simulation environment under various attacks. Such attacks compromise sensors, leading to inaccurate state observations and consequent malfunctions in TSC systems. When the sensors are intruded on or compromised, traffic disruptions could potentially occur, even leading to traffic safety accidents and substantial economic losses. To this end, it is crucial to develop innovative defense algorithms to mitigate the impact of sensor-related anomalies for TSC.

### 1.2. Challenges

Various solutions have been proposed in the context of online applications to address security issues faced by sensors (Sun et al., 2021). However, these methods typically only address a single type of attack. For instance, Zhang et al.(Zhang et al., 2021) proposed a method to address an adversarial attack, and it significantly decreases performance as noise levels increase, especially in complex, high-dimensional state environments (Yang et al., 2022). Lin et al. (Lin et al., 2017) employed model-based methods, e.g., a Multi-Layer Perceptron (MLP) for data prediction to address a data loss problem. Similarly, Mei et al. (Mei et al., 2023) advocated for interpolation techniques during training

---

[1]Beijing University of Posts and Telecommunications
[2]University of Technology Sydney [3]Lanzhou University. Correspondence to: Haipeng Peng <penghaipeng@bupt.edu.cn>.

*Proceedings of the 42nd International Conference on Machine Learning*, Vancouver, Canada. PMLR 267, 2025. Copyright 2025 by the author(s).

to solve a missing data problem. Yang et al. (Yang et al., 2023) proposed an offline method to solve the security problem of robotics. However, in a TSC system that requires real-time decision based on traffic volume, the above methods need to collect a large amount of data offline, which cannot meet the demand of real-time dynamic change of TSC data. The challenges faced by the existing TSC systems can be summarized as follows:

- Recent TSC algorithms demonstrate significant performance degradation when undergoing adversarial attacks or sensor damage, often resulting in traffic congestion, indicating a lack of security resilience.

- Existing defense methods address only one or two types of attacks and lack a comprehensive framework to address multiple security issues holistically.

- Current offline methods typically rely on the collection of large datasets to train static models, which struggle to make accurate decisions when faced with untrained real-time data.

### 1.3. Contributions

Diffusion models have recently achieved great results in image generation and RL control (Janner et al., 2022; Wang et al., 2022; Yang et al., 2023). The training of a diffusion model consists of the noise addition and denoising processes. The denoising process has the potential to evade noise attacks undergone by TSC systems. Moreover, the diffusion models have a strong data generation ability that has the potential to solve the missing data problem in TSC systems. Our contributions in addressing these challenges can be summarized as follows:

- We propose RobustLight, a framework to enhance the robustness of online TSC systems, consisting of a TSC agent and a dynamic state filling (DSI) agent. The DSI agent uses a model-free RL algorithm with an enhanced diffusion model to recover TSC data in real-time, ensuring optimal strategies without altering the original TSC algorithms. This architecture prevents data missing without changing the TSC algorithms.

- The RobustLight framework adopts two algorithms, denoise and repaint, which leverage the trained diffusion model of DSI to defend against adversarial attacks and handle missing data in TSC systems.

- Experiments show that RobustLight improves the average travel time of existing TSC algorithms by 50.43% under various adversarial attacks and missing data scenarios, with state recovery closely matching pre-attack distributions, demonstrating its enhanced robustness.

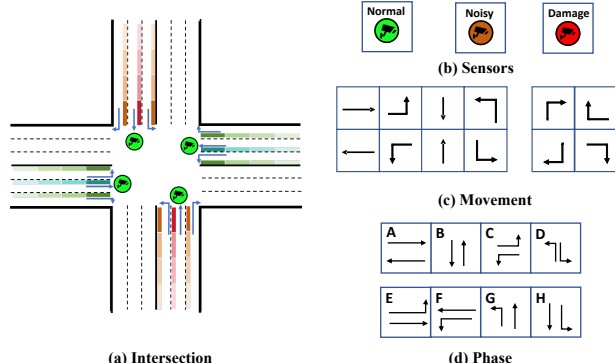

*Figure 1.* Definition of the TSC.

## 2. Preliminary

### 2.1. Definitions of TSC

We use a four-way intersection, as shown in Figure 1, to illustrate the concepts and summarize the definition of TSC.

**Intersection.** Each road network consists of multiple intersections, each with multiple, e.g., $N$, road segments, denoted by $(Inter_1, ..., Inter_N)$.

**Traffic perception.** Each intersection has four directional sensors (e.g., cameras, radars) monitoring vehicles in three lanes, with different security states indicated by colors: orange for noise attacks, red for sensor damage, and green for normal operation, as shown in Figure 1(b).

**Traffic movement.** The vehicle passes through the intersection from entering the lane ($lane_{in}$) to leaving the lane ($lane_{out}$). This traffic movement is represented as $TM = (lane_{in}, lane_{out})$, as shown in Figure 1(c).

**Traffic signal phase.** Two movements i.e., $(TM_i, TM_j, i \neq j)$ form a phase, represented as $p_w = (TM_i, TM_j)$, as depicted in Figure 1(d).

### 2.2. Adversarial Attack and Physical Sensor Damage

We introduce the concepts of four classic adversarial attacks and the definition of TSC sensor attacks.

**Gaussian noise attack.** The Gaussian noise attack adds Gaussian random noise $\mathcal{N}$ within the scale or intensity $k$ to the state $s$, represented as $\tilde{s}_t = s_t + k \cdot \mathcal{N}(\mu, \sigma^2)$

**U-rand attack.** The uniform random noise (U-rand) attack adds U-rand noise $\mathcal{U}$ within the intensity $k$ to the state $s$, i.e., $\tilde{s}_t = s_t + k \cdot \mathcal{U}(\mathcal{I}, \mathcal{I})$, where $I$ is the identity matrix.

**MAD attack.** The maximum action-difference (MAD) attack selects noise within a given range $k$ to maximize the difference between two policies $\pi_\phi(\cdot|s)$ and $\pi_\phi(\cdot|\tilde{s})$ in the policy space, denoted as $\tilde{s}_t = s_t + $

$\arg\max_{\tilde{s}\in\mathbf{B}_d(s,k)} D(\pi_\phi(\cdot|s) \parallel \pi_\phi(\cdot|\tilde{s}))$, where $\mathbf{B}_d(s,k)$ is the $\ell_\infty$ ball centered at state $s$ with radius $k$.

**MinQ attack.** The Minimum $Q$-value (adversarial) attack selects the minimum $Q$-value within a certain range $k$ and adds it as noise to the original state $s$, denoted as $\tilde{s}_t = s_t + \arg\min_{\tilde{s}\in\mathbf{B}_d(\tilde{s},k)} Q(\tilde{s}_t, \pi_\phi(\cdot|\tilde{s}))$.

**Physical sensor damage.** For each intersection, sensor damage due to weather or human factors causes state dimensions to be unobserved, represented as $\tilde{s}_t = Mask \cdot s_t$.

### 2.3. Diffusion Process and Guided Diffusion

The diffusion model (Ho et al., 2020b) is a probabilistic deep learning model that generates samples through diffusion and inverse diffusion processes.

**Forward process.** In this process, the data is deformed by introducing random noise, which makes the data gradually lose structured information and eventually turn into random noise. The forward diffusion chain gradually adds noise to the data $x_0$ (sampled from the distribution $q(x_0)$) over $T$ steps using a pre-defined variance schedule $\beta_t \{\beta_t \in (0,1)\}_{t=1}^{T}$, as given by

$$q\left(\mathbf{x}_t \mid \mathbf{x}_0\right) = \mathcal{N}\left(\mathbf{x}_t; \sqrt{\bar{\alpha}_t}\mathbf{x}_0, (1-\bar{\alpha}_t)\mathbf{I}\right), \quad (1)$$

where $\alpha_t = 1 - \beta_t$ and $\bar{\alpha}_t = \prod_{i=1}^{t} \alpha_i$. Equation (1) allows for the use of reparametrization (Gu et al., 2022) to directly obtain the noisy data $x_t$ corresponding to a specific timestep $t$ from the initial data without the need for multi-step iteration.

**Reverse process.** This process is assumed to follow a Gaussian distribution probability, as given by

$$p_\theta\left(\mathbf{x}_{t-1} \mid \mathbf{x}_t\right) = \mathcal{N}\left(\mathbf{x}_{t-1}; \boldsymbol{\mu}_\theta\left(\mathbf{x}_t, t\right), \boldsymbol{\Sigma}_\theta\left(\mathbf{x}_t, t\right)\right), \quad (2)$$

where $\boldsymbol{\mu}_\theta\left(\mathbf{x}_t, t\right)$ represents the mean of the Gaussian distribution that needs to be predicted by the neural network, and $\boldsymbol{\Sigma}_\theta\left(\mathbf{x}_t, t\right)$ is the pre-defined variance. After training, the neural network predicts the noise term. We sample data from an isotropic Gaussian noise and run the reverse diffusion process from $t = T$ to $t = 0$.

Guided Diffusion (Kim & Oh, 2022) extends diffusion by introducing external guidance during the reverse process. It controls the generation of specific content. Classifier-free guidance (Ho & Salimans, 2022) is a widely considered type of guided diffusion, which adds condition $c$ to the $\mu_\theta$ network in the diffusion reverse process, as given by

$$\boldsymbol{\mu}_\theta^{guided}(x_t, t, c) = \boldsymbol{\mu}_\theta(x_t, t) + \omega(\boldsymbol{\mu}_\theta(x_t, t, c) - \boldsymbol{\mu}_\theta(x_t, t)), \quad (3)$$

where $\omega$ is a weighting coefficient that controls the trade-off between conditional and unconditional generations.

## 3. RobustLight

In this section, we elaborate on the proposed RobustLight framework, which integrates a new dynamic state infilling (DSI) algorithm, and two new processes, namely, the no-attack training process and the attacked testing process, into any type of the existing TSC algorithms to protect from data false injection and missing data. Specifically, we propose DSI to train an improved diffusion model. The denoise and repaint algorithms use the trained diffusion model of DSI to solve adversarial attacks and missing data problems. The integration of these algorithms improve the robustness of TSC in unreliable real-time traffic environments.

### 3.1. TSC Algorithm

A TSC algorithm implements the basic functions of signal control, serving as a TSC agent to interact with the traffic environment. This allows for the use of various TSC algorithms, including both traditional and RL-based algorithms.

In what follows, we elaborate on the TSC algorithms (namely, MaxPressure (Cools et al., 2013), Advanced-Maxpressure (Zhang et al., 2022b), Colight (Wei et al., 2019a), Advanced-Colight (Zhang et al., 2022b), Mplight (Chen et al., 2020), and Advanced-Mplight (Zhang et al., 2022b)). The specific definitions of state, action, and reward are given by

• **State:** The Efficient Pressure (EP) (Wu et al., 2021) and Running Vehicle (RV) are the input state $s$.

• **Action:** The traffic signal phase is action $a$.

• **Reward:** The negative of the queue length is the reward $r$.

We represent the TSC state trajectory $\tau$ as a sequence, as given by

$$\tau_t^s = \{s_1, s_2, ..., s_{t-1}, s_t\}, \quad (4)$$

where $t$ indicates the RL timestep. We can update the RL-based TSC agent according to the Bellman function. Then, the Q-value in the RL-based TSC evolves as

$$Q(s,a) \longleftarrow Q(s,a) + \alpha \left[r + \gamma \max_{a'\in a} Q'(s',a') - Q(s,a)\right], \quad (5)$$

where $\alpha$ represents the learning rate, $\gamma$ is the discount factor, and $Q'(s',a')$ denotes the target Q-value for the next state-action pair. For the TSC algorithms based on RL, a replay buffer is used to store tuples $(s,a,r,s')$.

### 3.2. Dynamic State Infilling (DSI) Algorithm

We design a DSI algorithm using an improved diffusion model as the policy to reduce noise in online model-free RL. The TSC agent's replay buffer is used to update the policy and adapt to real-time environments. The state, action, and reward of the DSI agent are defined as follows:

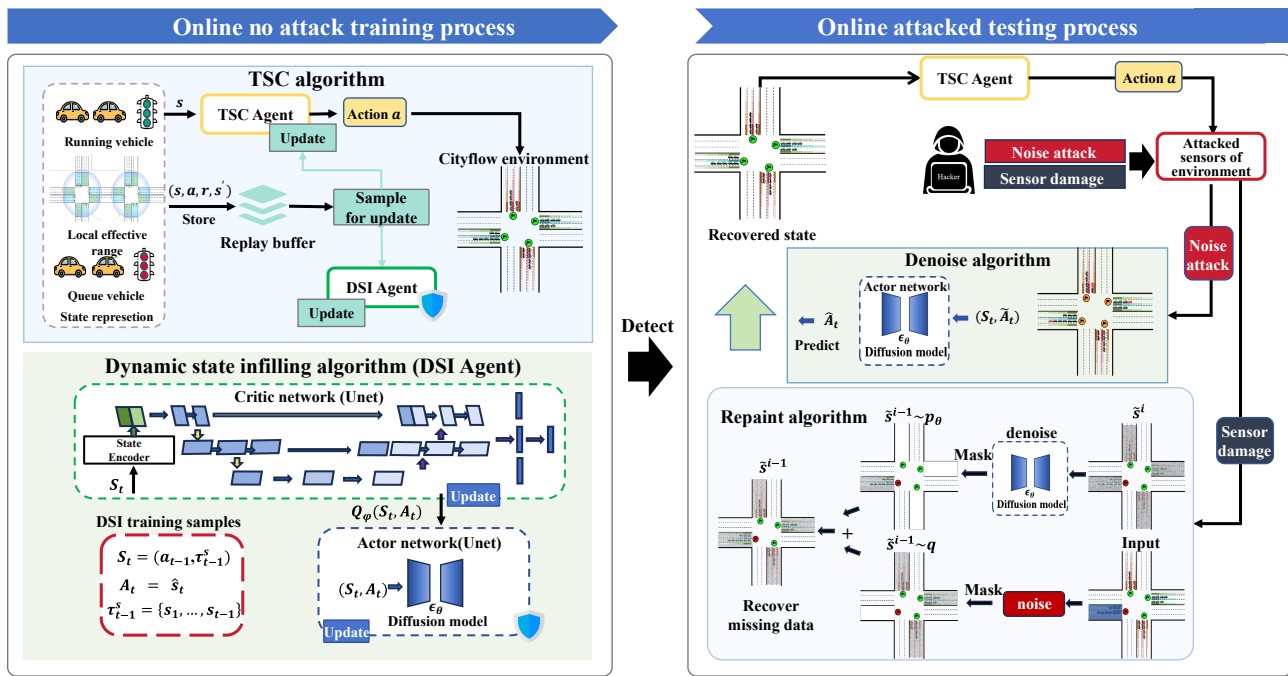

*Figure 2.* RobustLight uses a two-process framework with four algorithms: the TSC and DSI algorithms train on normal data to enhance efficiency and robustness, while the Denoise and Repaint algorithms mitigate noise and infer missing data during attack testing.

• **State.** We represent the DSI agent's state as $S_t = (a_{t-1}, \tau_{t-1}^s)$, where $a_{t-1}$ is the previous TSC action and $\tau_{t-1}^s = \{s_1, ..., s_{t-1}\}$ is the TSC state trajectory.

• **Action.** The DSI denoises the noisy or missing state $\tilde{s}_t$ to the recovered state $\hat{s}_t$. Hence, the action at time $t$ can be represented as $\hat{A}_t = \hat{s}_t$.

• **Reward.** The reward is the negative of the queue length. The DSI agent consists of an actor and a critic network. **Critic network.** In the critic network parameterized by $\varphi$, a U-Net architecture (Ronneberger et al., 2015) identical to that of the actor network is employed. The final layer of the critic network produces the $Q$ value. The network is updated using the Bellman error, as given by

$$L_\varphi = r + \lambda Q_\varphi(S_{t+1}, A_{t+1}) - Q_\varphi(S_t, A_t), \quad (6)$$

where $\lambda$ is the discount factor.

**Actor network.** For the actor network, we use action gradient ascent to update the control policy (Yang et al., 2023)

$$A_t = A_t + \eta \nabla_A Q_\varphi(\mathbf{S}_t, A_t), \quad (7)$$

where $\eta$ is the learning rate of the action gradient ascent. We employ an improved diffusion model to denoise and predict the original state $\hat{A}_t = \hat{s}_t$ through the reverse process:

$$\hat{A}_t \sim p_\theta(\tilde{A}_t^{0:k}|S_t) = f_k(\tilde{A}_t) \prod_{i=1}^k p_\theta(\tilde{A}_t^{i-1}|\tilde{A}_t^i, S_{t-1}), \quad (8)$$

where $f_k(\tilde{A}_t) = \sqrt{\bar{\alpha_k}}\tilde{A}_t$, and $\prod_{i=1}^k p_\theta(\tilde{A}_t^{i-1}|\tilde{A}_t^i, S_{t-1})$ can be modeled as Gaussian distribution, as follows:

$$\mathcal{N}\left(\tilde{\boldsymbol{A}}_t^{i-1}; \boldsymbol{\mu}_\theta\left(\tilde{\boldsymbol{A}}_t^i, \boldsymbol{S}_{t-1}, i\right), \boldsymbol{\Sigma}_\theta\left(\tilde{\boldsymbol{A}}_t^i, \boldsymbol{S}_{t-1}, i\right)\right). \quad (9)$$

According to (Ho et al., 2020b), the denoising process for each step in the diffusion is expressed as

$$\tilde{A}_t^{i-1}|\tilde{A}_t^i = \frac{\tilde{A}_t^i}{\sqrt{\alpha_i}} - \frac{\beta_i}{\sqrt{\alpha_i(1-\bar{\alpha}_i)}}\epsilon_\theta(\tilde{A}_t^i, S_t, i) + \sqrt{\tilde{\beta}_i}\epsilon. \quad (10)$$

To handle denoising tasks with improved diffusion, we use the beta schedule (Zhihe & Xu) $\beta_i = 1 - \alpha_i = e^{-\frac{b}{i+a+c}}, \bar{\alpha}_k = \prod_{i=1}^k \alpha_i$, and $\tilde{\beta}_i = \frac{1-\bar{\alpha}_{i-1}}{1-\bar{\alpha}_i}\beta_i$ to address small to medium-scale noises, where $\epsilon_\theta$ is the noise network. Unlike other beta (Xiao et al., 2021a; Nichol & Dhariwal, 2021a; Ho et al., 2020a) schedules that generate data from pure noise, our approach employs the U-Net architecture and a conditional generative diffusion model for noise prediction.

**Lemma 3.1.** *The lemma basis of diffusion for denoising is to minimize the sum entropy of the denoised data:*

$$L_\theta = \sum_{t=1}^T \mathbb{E}q_{(A_t)}[-\log p_\theta(\tilde{A}_t^0|S_{t-1})], \quad (11)$$

*which can be optimized using the variational lower bound*

---

**Algorithm 1** The training process of RobustLight

**Initialize:** DSI algorithm $Q_\varphi$ critic network, $Q_\theta$ actor network, Replay buffer $\mathcal{R}$ and target network $Q_{\varphi^-}, Q_{\theta^-}$.

1: **for** $j = 1$ to $E$ **do**
2:     Get init state $s_0$ from Cityflow
3:     **for** $t = 1$ to $T$ **do**
4:         Execute TSC algorithms, including Colight, Advanced-Colight, Maxpressure,..., etc.
5:         Training the TSC agent by Equation 5.
6:         Sample batch $S = (\tau^s, a)$ and $A = s$ from $\mathcal{R}$.
7:         Update critic network $Q_\varphi$ by Equation 6.
8:         **for** $n = 1$ to $N$ **do**
9:           Execute gradient ascend by Equation 7.
10:         **end for**
11:         Update actor network $Q_\theta$ by Equation 12.
12:         Update target critic network by:
13:         $\varphi^- = \eta\varphi + (1-\eta)\varphi^-$
14:         Update target actor network by
15:         $\theta^- = \eta\theta + (1-\eta)\theta^-$
16:     **end for**
17: **end for**
18: Return Actor network $Q_\theta$

---

**Algorithm 2** Repaint algorithm of RobustLight

1: Input $\tilde{A}_t^i, S_{t-1}, m$
2: **for** $i = 1$ to $K$ **do**
3:     **for** $u = 1$ to $U$ **do**
4:       $\epsilon \sim \mathcal{N}(0, I)$ if $i > 1$, else $\epsilon = 0$
5:       Get $\tilde{A}_{t,known}^{i-1}$ by Equation (13)
6:       $z \sim \mathcal{N}(0, I)$ if $i > 1$, else $z = 0$
7:       Get $\tilde{A}_{t,unknown}^{i-1}$ by Equation (14)
8:       Get recovered $\tilde{A}_t^{i-1}$ by Equation (15)
9:       **if** $u < U$ and $i > 1$ **then**
10:         $\tilde{A}_t^i \sim \mathcal{N}(\sqrt{1 - \beta_{i-1}}\tilde{A}_t^{i-1}, \beta_{i-1}I)$
11:       **end if**
12:     **end for**
13: **end for**
14: Return $\hat{A}_t = \tilde{A}_t^0$

---

sion timestep. We perform the denoising process in each $i$ timesteps to predict the original state $\hat{s}_t^0$ using Equation (8).

### 3.4. Repaint Algorithm

The concept of utilizing diffusion for image repaint is inspired by (Lugmayr et al., 2022). The core idea is to use the known sensor data to infer the unknown sensor data. We adapt and refine the image repaint process to the TSC and introduce the repaint algorithm to interpolate the damaged or missing TSC state, as illustrated in Figure 2.

We use a well-trained conditional denoising diffusion probabilistic model, based on DSI (distinct from (Zhihe & Xu)), to effectively restore the original state $\hat{A} = \hat{s}$, where the unknown part is represented by $m \odot \tilde{A}^{i-1}$ with $m$ being the mask matrix, and the known part is $(1 - m) \odot \tilde{A}^{i-1}$ stands for unmask matrix. We note that the reverse process of diffusion from $\tilde{A}^i$ to $\tilde{A}^{i-1}$ depends solely on $\tilde{A}^i$, as long as we maintain the correctness properties of the corresponding distribution. Therefore, we update the known state $(1 - m) \odot \tilde{A}^{i-1}$. According to Equation (1), we sample the known state at any diffusion timestep $i$, and use Equation (2) for the unknown state. We use the following expression for one reverse step:

$$\tilde{A}_{t,known}^{i-1} = \sqrt{\bar{\alpha}_i}\tilde{A}_{t,known} + \sqrt{1 - \bar{\alpha}_i}\epsilon, \quad (13)$$

$$\tilde{A}_{t,unknown}^{i-1} = \frac{1}{\sqrt{\alpha_i}}\tilde{A}_{t,unknown}^i - \frac{\beta_i}{\sqrt{\alpha_i(1 - \bar{\alpha}_i)}}$$
$$\epsilon_\theta(\tilde{A}_t^i, S_{t-1}, i) + \sqrt{\beta_i}z, \quad (14)$$

$$\tilde{A}_t^{i-1} = m \odot \tilde{A}_{t,known}^{i-1} + (1 - m) \odot \tilde{A}_{t,unknown}^{i-1}. \quad (15)$$

Thus, $\tilde{A}_{known}^{i-1}$ is sampled using the known state in the given state $m \odot A^0$, and $\tilde{A}_{unknown}^{i-1}$ is sampled from the condition model given the previous iteration $\tilde{A}^i$. These are then combined into the new sample $\tilde{A}^{i-1}$. The repaint algorithm is summarized in Algorithm 2.

---

*(VLB). For the detailed derivation process, please refer to (Zhihe & Xu; Ho et al., 2020a).*

Expanding $L_\theta$, we obtain the following Non-Markov loss to update the actor network:

$$L_\theta = \mathbb{E}_{i \sim \mathcal{U}_K, \epsilon_t \sim \mathcal{N}(0,I), (A_{t-N}, ..., A_{t+M-1}) \in D_\nu}$$
$$\left\| \epsilon_\theta(\tilde{A}_t^i, S_{t-1}, i) - \epsilon_t^i \right\|_2 + \sum_{m=t+1}^{t+M-1} \left\| \epsilon_\theta(\tilde{A}_m^i, \hat{S}_{m-1}, i) - \epsilon_m^i \right\|_2,$$
$$(12)$$

where $\hat{S}_{m-1} = (a_{m-1}, \tau_{m-1}^{\hat{s}})$, and $\tau_{m-1}^{\hat{s}}$ is the predicted state trajectory. This loss function $L_\theta$ balances the trade-off between the current RL timestep and future RL timesteps, aiming to minimize the accumulated error over time. Detailed training algorithm is summarized in Algorithm 1.

### 3.3. Denoise Algorithm

In the testing phase, we may unintentionally expose the sensors within the TSC algorithm to various attacks, including Gaussian, MAD, U-rand, and MinQ attacks.

In the denoise algorithm, we use the trained actor diffusion model from the DSI algorithm to recover the original TSC state $\hat{A}_t = \hat{s}_t$. We use the diffusion reverse process to denoise. As described in Equation (10), we input $(S_t, \tilde{A}_t)$ and the diffusion time step $i$ into the actor diffusion network. In each diffusion timestep, the data currently under a noise attack is denoised to obtain the data for the next diffu-

*Table 1.* **Performance of ATT in JiNan, HangZhou. "-" implies that traditional algorithms are not adapted to MinQ and MAD attacks. Our RobustLight recovers the state of traditional and RL-based TSC algorithms to evaluate the performance.**

| Dataset | Noise Type | Noise Scale | FixedTime | MaxPressure | | Advanced-MaxPressure | | Advanced-MpLight | | CoLight | | Advanced-CoLight | |
|---|---|---|---|---|---|---|---|---|---|---|---|---|---|
| | | | base | base | RobustLight | base | RobustLight | base | RobustLight | base | RobustLight | base | RobustLight |
| $JiNan_1$ | Gaussian | 3.5 | 428.11±0.00 | 301.97±2.14 | **296.01±2.18** | 285.43±1.82 | **283.48±1.22** | 356.86±19.78 | **302.16±2.86** | 287.76±1.52 | **282.35±1.96** | 320.57±2.82 | **294.57±2.02** |
| | | 4.0 | | 305.88±2.92 | **298.95±2.69** | 290.28±2.17 | **289.71±1.59** | 377.91±38.59 | **311.59±4.69** | 314.93±28.28 | **288.07±2.27** | 338.93±15.32 | **298.23±5.16** |
| | U-rand | 3.5 | | 325.09±2.65 | **319.01±2.17** | 312.27±2.92 | **306.12±1.79** | 538.66±132.69 | **349.89±22.56** | 407.41±24.57 | **312.27±4.07** | 449.13±21.14 | **358.22±7.02** |
| | | 4.0 | | 331.58±3.57 | **322.93±2.67** | 319.18±2.91 | **311.32±1.97** | 549.41±120.08 | **364.39±28.71** | 352.63±7.85 | **320.48±3.29** | 460.46±22.41 | **360.82±4.79** |
| | MAD | 3.5 | | - | - | - | - | 321.24±14.89 | **280.45±2.99** | 487.69±50.75 | **277.37±1.16** | 479.63±8.82 | **283.13±1.56** |
| | | 4.0 | | - | - | - | - | 338.14±24.71 | **287.04±2.69** | 555.71±64.93 | **279.15±1.22** | 520.09±7.12 | **288.07±2.27** |
| | MinQ | 3.5 | | - | - | - | - | 313.73±11.87 | **291.58±8.12** | 683.38±94.91 | **277.93±3.47** | 394.39±50.27 | **323.25±20.54** |
| | | 4.0 | | - | - | - | - | 321.55±12.64 | **295.12±6.06** | 716.58±134.92 | **281.13±1.96** | 394.63±47.37 | **331.52±10.64** |
| $JiNan_3$ | Gaussian | 3.5 | 383.01±0.00 | 275.32±1.24 | **270.16±1.70** | 267.76±0.93 | **263.81±0.94** | 383.41±58.34 | **272.12±2.25** | 289.03±10.62 | **258.33±2.06** | 410.96±132.05 | **288.87±3.59** |
| | | 4.0 | | 279.69±1.92 | **274.18±1.25** | 272.17±1.26 | **271.87±2.03** | 550.13±188.98 | **279.86±2.31** | 293.55±1.86 | **261.63±1.66** | 461.57±126.58 | **294.84±5.45** |
| | U-rand | 3.5 | | 299.34±2.22 | **288.71±2.56** | 293.46±1.22 | **289.66±2.09** | 454.32±138.56 | **307.59±20.82** | 407.89±50.61 | **300.36±7.67** | 987.04±42.62 | **561.02±62.23** |
| | | 4.0 | | 306.02±2.36 | **292.13±2.65** | 298.58±2.19 | **294.64±2.42** | 500.65±122.68 | **313.02±17.33** | 554.51±80.09 | **306.03±6.59** | 1015.29±41.75 | **596.24±67.48** |
| | MAD | 3.5 | | - | - | - | - | 498.88±208.66 | **259.84±1.52** | 534.43±87.67 | **263.73±2.57** | 431.77±17.35 | **260.78±1.42** |
| | | 4.0 | | - | - | - | - | 594.48±218.45 | **264.52±1.38** | 550.26±93.93 | **267.03±2.09** | 474.16±17.21 | **260.78±1.42** |
| | MinQ | 3.5 | | - | - | - | - | 652.05±230.01 | **283.01±34.19** | 726.85±282.56 | **267.34±3.65** | 515.34±167.83 | **309.65±9.99** |
| | | 4.0 | | - | - | - | - | 733.62±233.21 | **295.12±57.76** | 744.21±274.72 | **273.13±4.52** | 516.23±167.39 | **329.81±14.76** |
| $HangZhou_1$ | Gaussian | 3.5 | 495.57±0.00 | 332.03±2.01 | **322.45±1.25** | 327.37±1.94 | **324.84±1.79** | 564.65±103.29 | **351.68±40.82** | 356.33±6.38 | **322.62±5.96** | 480.38±22.93 | **327.98±2.45** |
| | | 4.0 | | 335.97±2.49 | **326.58±1.36** | 331.93±1.57 | **329.25±1.21** | 490.72±93.84 | **366.62±50.39** | 371.97±12.44 | **337.58±5.68** | 510.38±25.17 | **331.21±3.65** |
| | U-rand | 3.5 | | 356.53±3.72 | **338.42±1.74** | 354.91±3.31 | **297.34±1.39** | 373.88±34.05 | **328.18±3.17** | 647.64±54.89 | **435.33±20.02** | 717.12±70.08 | **473.85±32.68** |
| | | 4.0 | | 361.58±4.37 | **341.16±2.21** | 359.61±3.34 | **350.86±3.12** | 384.75±42.24 | **330.47±3.11** | 475.49±29.78 | **441.33±17.05** | 738.51±63.18 | **487.74±31.44** |
| | MAD | 3.5 | | - | - | - | - | 803.38±92.02 | **325.49±28.86** | 402.67±17.42 | **324.55±6.63** | 463.64±33.44 | **309.24±0.94** |
| | | 4.0 | | - | - | - | - | 782.23±121.05 | **320.91±2.73** | 419.07±18.26 | **331.47±8.62** | 491.64±39.78 | **352.58±3.71** |
| | MinQ | 3.5 | | - | - | - | - | 778.04±78.91 | **420.46±87.75** | 466.14±70.37 | **317.36±3.96** | 449.54±8.22 | **406.32±6.41** |
| | | 4.0 | | - | - | - | - | 764.88±78.52 | **435.53±69.87** | 460.29±62.45 | **331.93±3.91** | 482.98±7.29 | **430.53±8.49** |

## 4. Experiments

We conduct experiments with real-world datasets to evaluate RobustLight's generalization and efficiency on NVIDIA P100 hardware, using the hyperparameters in Table 10.

### 4.1. Datasets

We use real-world traffic flow and road topology datasets for our experiments, with Cityflow (Zhang et al., 2019) as the simulator to evaluate Average Travel Time (ATT) and exit points with a simulation time of 60 minutes for all vehicles. The datasets include vehicle start and end points, following a fixed motion model. Seven traffic datasets from three cities JiNan and HangZhou (China) and New York (Zheng et al., 2019) (USA) are used.

**JiNan Datasets:** The JiNan road network consists of 12 intersections (in a $3 \times 4$ grid). It includes three traffic flow datasets: $JiNan_1$, $JiNan_2$, and $JiNan_3$.

**HangZhou Datasets:** The HangZhou network encompasses 16 intersections (in a $4 \times 4$ grid) and features two datasets: $HangZhou_1$ and $HangZhou_2$.

**New York Datasets:** The New York network features a more complex structure with 192 intersections ($28 \times 7$ grid) and includes two datasets: $Newyork_1$ and $Newyork_2$.

### 4.2. Compared Methods

**Traditional Methods:** These methods include Fixed-Time (Webster, 1958), which uses a fixed green phase time;

Advanced-Maxpressure (Zhang et al., 2022a), which uses running and waiting vehicles to choose the phase; and Maxpressure (Gershenson, 2004), which uses waiting vehicles to choose the phase.

**RL-based Methods:** For RL benchmarks, we consider Co-Light (Wei et al., 2019a), which uses waiting and neighboring vehicles to select the signal phase; Advanced-CoLight (Zhang et al., 2022a), which employs waiting and running vehicles with a graph attention neural network; and Advanced-Mplight (Zhang et al., 2022a), which uses the FRAP (Zheng et al., 2019) model for signal phase selection.

**RobustLight Method:** RobustLight integrates the base methods of traditional and RL-based TSC algorithms to recover data in real-time, and then evaluates the ATT under different sensor noise attacks and sensor damage. Results are presented as the average of ten independent runs.

### 4.3. Results

This subsection presents the results of our experiments, evaluating RobustLight's performance under various conditions, including resilience to noise attacks and sensor damage, using ATT on real-world traffic datasets.

#### 4.3.1. NOISE ATTACK ON STATE RESULTS

Table 15 summarizes our experimental results for the $JiNan$ and $HangZhou$ datasets based on the ATT metric. The noise scale range is based on the value of $k$ for state noise attacks, as described in Section 2.2. The re-

*Table 2.* **ATT in JiNan and HangZhou: 25% refers to missing data in $sensor_W$, and 50% refers to $sensor_W$ and $sensor_E$.**

| Dataset | Mask Scale | FixedTime | MaxPressure | | Advanced-MaxPressure | | Advanced-MpLight | | CoLight | | Advanced-CoLight | |
|---|---|---|---|---|---|---|---|---|---|---|---|---|
| | | base | base | RobustLight | base | RobustLight | base | RobustLight | base | RobustLight | base | RobustLight |
| JiNan1 | 25% | 428.11±0.00 | 386.74±0.00 | **324.31±22.74** | 353.04±0.00 | **302.84±9.43** | 398.27±86.93 | **337.63±50.23** | 400.93±19.66 | **372.93±7.43** | 326.14±16.52 | **298.35±16.32** |
| | 50% | | 798.90±0.00 | **613.23±108.81** | 1061.92±0.00 | **548.84±152.33** | 1052.83±102.63 | **642.62±111.28** | 849.61±77.93 | **766.91±81.83** | 732.37±52.93 | **680.37±97.32** |
| JiNan2 | 25% | 368.76±0.00 | 272.51±0.00 | **273.78±3.61** | 323.13±0.00 | **274.84±4.74** | 701.85±200.84 | **259.36±6.43** | 637.94±285.96 | **308.56±22.59** | 290.86±24.79 | **259.56±7.59** |
| | 50% | | 836.81±0.00 | **748.23±132.12** | 1209.97±0.00 | **751.53±268.33** | 985.42±64.04 | **665.92±148.31** | 880.23±52.93 | **754.23±173.95** | 725.57±57.89 | **639.94±100.49** |
| JiNan3 | 25% | 383.01±0.00 | 289.81±0.00 | **288.74±9.83** | 340.81±0.00 | **286.54±10.85** | 824.35±228.64 | **368.62±104.37** | 367.94±9.36 | **315.43±9.67** | 363.84±54.43 | **301.63±38.26** |
| | 50% | | 823.48±0.00 | **574.97±69.0** | 1109.57±0.00 | **592.56±222.45** | 947.84±150.34 | **567.37±75.16** | 816.73±56.68 | **382.39±37.83** | 756.37±162.69 | **407.68±78.39** |
| HangZhou1 | 25% | 495.57±0.00 | 369.77±0.00 | **350.8±20.22** | 513.15±0.00 | **371.64±28.35** | 510.37±92.75 | **372.12±21.49** | 490.36±13.84 | **424.36±13.22** | 401.87±48.31 | **328.26±12.67** |
| | 50% | | 722.43±0.00 | **714.32±76.93** | 1186.56±0.00 | **752.84±399.84** | 1170.97±71.83 | **515.77±180.92** | 786.32±40.86 | **542.91±108.24** | 842.29±235.36 | **541.56±122.96** |
| HangZhou2 | 25% | 406.65±0.00 | 372.12±0.00 | **353.85±3.66** | 405.27±0.00 | **356.64±4.64** | 362.93±15.36 | **350.26±10.62** | 397.88±20.68 | **390.43±10,37** | 378.86±26.86 | **375.64±22.32** |
| | 50% | | 533.81±0.00 | **447.91±70.34** | 781.57±0.00 | **542.37±161.07** | 639.52±127.45 | **522.92±157.32** | 663.72±80.82 | **470.91±34.82** | 508.93±30.27 | **444.37±61.24** |

sults reveal that the incorporation of Robustlight leads to improved performance across all methods, demonstrating its effectiveness in mitigating various types of noise attacks.

### 4.3.2. SENSOR DAMAGE ON STATE RESULTS

We compare traditional and RL-based TSC algorithms, focusing on deliberate sensor attacks on $sensor_W$ and $sensor_E$, with data from these directions masked to simulate damage. We use the Repaint algorithm within RobustLight for data completion, allowing us to evaluate the TSC algorithm's performance before and after this process, as shown in Table 2. Our findings lead to several key conclusions. Under noisy conditions, MaxPressure outperforms Advanced-MaxPressure with an 11.6% average improvement across all datasets. Advanced-CoLight surpasses CoLight with an 18.4% improvement. The RobustLight algorithm enhances performance in nearly all methods, with Advanced-CoLight showing a 10.3% improvement. Following sensor damage in $sensor_W$ and $sensor_E$, all methods perform worse than FixedTime, but our algorithms effectively recover missing data and still outperform FixedTime with damage in one direction, highlighting the importance of addressing sensor damage in real-world TSC deployments.

*Table 3.* **Performance comparison with other benchmarks.**

| Dataset | Noise Type | Noise Scale | Advanced-CoLight | | | |
|---|---|---|---|---|---|---|
| | | | base | RobustLight | Diffusion Linear-Beta | LSTM |
| $JiNan_1$ | Gaussian | 3.5 | 320.57±2.82 | **294.57±2.02** | 324.04±3.04 | 892.0±213.7 |
| | | 4.0 | 338.93±15.32 | **298.23±5.16** | 326.83±6.56 | 914.18±215.7 |
| | Sensor | 25% | 326.14±16.52 | **298.35±16.32** | 350.96±17.32 | 1035.39±156.69 |
| | Damage | 50% | 732.37±52.93 | **680.37±97.32** | 689.13±98.43 | 1088.77±160.11 |
| $HangZhou_1$ | Gaussian | 3.5 | 480.38±22.93 | **327.98±2.45** | 480.53±33.55 | 1123.4±341.28 |
| | | 4.0 | 510.38±25.17 | **331.21±3.65** | 510.16±22.34 | 1140.26±325.02 |
| | Sensor | 25% | 401.87±48.31 | **328.26±12.67** | 386.13±29.28 | 1581.35±73.21 |
| | Damage | 50% | 842.29±235.36 | **541.56±122.96** | 551.44±40.16 | 1638.32±14.1 |

### 4.3.3. OTHER METHODS COMPARISON EXPERIMENTS

We compare the native Diffusion model with three Beta schedule methods (Xiao et al., 2021b; Ho et al., 2020b; Nichol & Dhariwal, 2021b) and LSTM model (Sun et al., 2021) on real-world datasets. As shown in Table 3, our RobustLight outperforms the native Diffusion model in all noise attack scenarios, demonstrating its effectiveness.

*Table 4.* Performance Comparison with DiffLight

| Dataset | Noise/Mask | Method | RobustLight | PSNR | MAE | ATT |
|---|---|---|---|---|---|---|
| JiNan1 | U-rand Noise (3.5) | DiffLight | No | 6.71 | 6.26 | 310.92 |
| | | Advanced-MaxPressure | Yes | **7.20** | **5.42** | **304.34** |
| HangZhou1 | U-rand Noise (3.5) | DiffLight | No | 6.85 | 7.93 | 361.31 |
| | | Advanced-MaxPressure | Yes | **7.71** | **5.12** | **297.34** |
| JiNan1 | Sensor Damage (25%) | DiffLight | No | 7.66 | 1.05 | 366.05 |
| | | Advanced-Colight | Yes | **9.34** | **0.89** | **304.13** |
| HangZhou1 | Sensor Damage (25%) | DiffLight | No | 18.05 | 1.84 | 372.53 |
| | | Advanced-Colight | Yes | **22.96** | 1.17 | **306.56** |

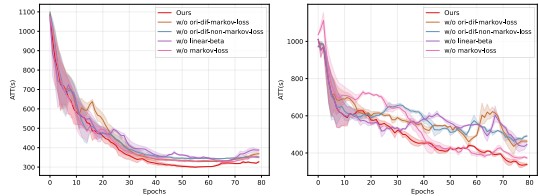

*Figure 3.* **Ablation of RobustLight based on Advanced-Colight in $JiNan_1$ and $HangZhou_1$ under Gaussian noise (scale 3.5).**

To validate our approach with DiffLight (Chen et al., 2024), we setup involved randomly masking data from Kriging Missing (12.5%, single-intersection-sensor failure) and Random Missing (12.5%, full-intersection failure). As demonstrated in Table 4, our method effectively addresses data missing scenarios (Kriging and random missing) while also exhibiting robust performance under noisy data conditions.

### 4.3.4. ABLATION EXPERIMENTS

We conduct ablation experiments on RobustLight to assess the impact of each component. "w/o linear beta" refers to the experiment without the linear beta schedule, "w/o markov-loss" excludes the Markov loss, "w/o ori-dif-markov-loss" omits the original diffusion model with Markov loss, and "w/o ori-dif-non-markov-loss" excludes the original diffusion model with non-Markov loss. Showing the effectiveness of RobustLight.

### 4.3.5. ROBUSTNESS RECOVERY ANALYSIS

We analyze RobustLight's effectiveness in recovering the original data using t-SNE (Van der Maaten & Hinton, 2008) plots. In the plots, yellow represents the original state distribution, blue represents the recovered states, and red shows

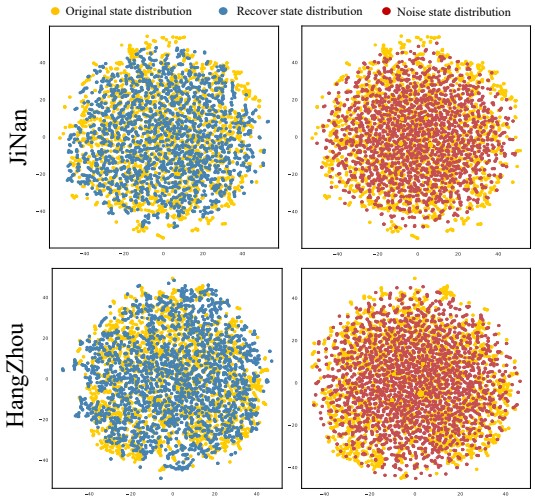

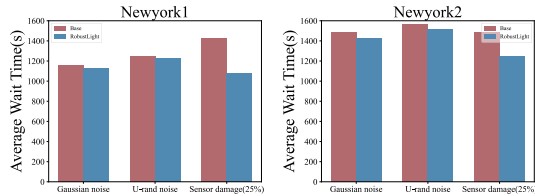

*Figure 6.* **Performance of RobustLight based on Advanced-Maxpressure in** $Newyork$ **transfer by** $JiNan_1$.

*Figure 4.* **State visulization of RobustLight based on Advanced-Colight in** $HangZhou_1$ **and** $JiNan_1$.

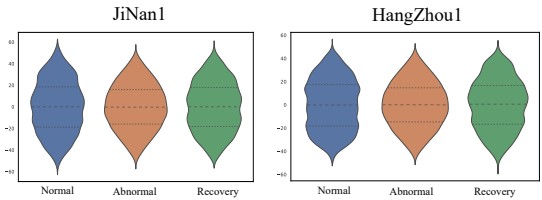

*Figure 5.* **Violin figure of normal, noise and recovery data.**

the distribution with Gaussian noise (scale 3.5). Figure 4 shows that the blue distribution is closer to the original, demonstrating RobustLight's strong robustness. Additionally, violin plots in Figure 5 show that the recovered data distribution closely matches the normal data.

To directly evaluate the recovery effect, we define the metric $E_{denoise} = \frac{1}{N} \sum_1^T |\hat{s} - s|$. Where $N$ is the number of intersections, $T$ is the running count, $\hat{s}$ is the recovered state, and $s$ represents the original state. is the original state. This metric measures the ability to restore data by the absolute difference between the original and reconstructed states. As shown in Table 5, RobustLight exhibits strong recovery performance.

### 4.3.6. MODEL GENERALIZATION

To assess generalization, we train a diffusion model based on the Advanced-Maxpressure algorithm using the $JiNan_1$

dataset and test it on the $Newyork$ dataset. As shown in Figure 6, transferring the model to $Newyork$ effectively mitigates noise attacks and demonstrates strong state restoration in a different urban traffic environment.

## 5. Discussion

**Q1. Is there any challenge to apply the diffusion model?** We evaluate the time consumption of the DSI algorithm (Table 17) to assess its computational costs and impact on TSC algorithms. As the noise range increases, more diffusion steps are needed, resulting in longer recovery times. A key challenge in large-scale scenarios is slow inference, driven by both the diffusion steps and the increasing number of intersections. To address this, we plan to reduce diffusion steps using DDIM and divide intersections into smaller sub-regions, each with its own actor diffusion model, to speed up inference. This discussion highlights the trade-off between accuracy and speed in diffusion models for real-time TSC systems and suggests ways to improve efficiency in large-scale deployments.

**Q2. Is there any report about cyber attacks on traffic control systems?** Traffic sensor attacks (Chen et al., 2019; Chowdhury et al., 2023a; Laszka et al., 2016a), demonstrated at DEFCON 22 by Cesar Cerrudo, highlight vulnerabilities in traffic control systems, where hackers target street magnetic sensors. Discussions on platforms like Quora ("hack traffic lights") further emphasize concerns about sensor safety. As AI systems become more prominent, ensuring their security is crucial, especially given the additional risks from extreme weather interference.

**Q3. How to detect noise and missing data?** To detect noise and missing data in our study, we use a combination of attack detection models and statistical methods. We assume that attacks, caused by factors such as weather conditions and radio interference, have been pre-detected, with the attack detection model identifying noisy or missing data based on abnormal sensor readings. Noise is detected through outlier detection and variance analysis, while missing data is identified by checking for null or NaN values and analyzing missingness patterns across sensors. Once detected, we handle noise using denoise algorithm and missing data through repaint algorithm of RobustLight, ensuring data

*Table 5.* $E_{denoise}$ **the smaller the value, the better.**

| Dataset | Noise Type | Noise Scale | CoLight | | Advanced-CoLight | |
|---|---|---|---|---|---|---|
| | | | base | RobustLight | base | RobustLight |
| $JiNan_1$ | Gaussian | 3.5 | 399.15±20.31 | **284.46±21.26** | 808.76±16.84 | **484.15±13.26** |
| | U-rand | 3.5 | 899.85±15.23 | **628.36±13.82** | 1814.56±31.36 | **1186.43±24.31** |
| $HangZhou_1$ | Gaussian | 3.5 | 399.15±21.35 | **284.30±16.37** | 1077.09±25.82 | **320.57±13.82** |
| | U-rand | 3.5 | 1198.21±13.37 | **535.67±16.31** | 2417.44±19.96 | **1202.61±38.74** |

*Table 6.* **Detection Rate and Throughput Comparison**

| Detection Rate | Dataset | Base | ATT | Base | Throughput |
|---|---|---|---|---|---|
| 80% | $JiNan_1$ | 487 | **297** | 5812 | **6154** |
| | $HangZhou_1$ | 463 | **326** | 2888 | **2938** |
| 60% | $JiNan_1$ | 487 | **328** | 5812 | **6131** |
| | $HangZhou_1$ | 463 | **331** | 2888 | **2930** |

integrity and robustness in the presence of environmental disturbances.

**Q4. If the RobustLight rely on the accuracy of detection?** To evaluate its reliance on detection accuracy, we integrated TP-FDS (Sarteshnizi et al., 2023), a method that detects anomalies by comparing new data distributions with historical data from the same period, achieving an AUC of 96% and an F1 score of 76%. TP-FDS identifies anomalies through multi-sensor cross-referencing (e.g., cameras and radar) or rule-based methods, such as detecting a queue increase from 3 to 5 during a north-south green light. Minor fluctuations, like queue changes from 3 to 5, minimally impact system efficiency. To assess RobustLight's dependence on detection accuracy, we conducted experiments simulating real-world scenarios with detection rates of 80% and 60%, as shown in Table 6. The results show that RobustLight has a robust performance in most detection rates.

**Q5. What are the practical considerations and potential barriers to real-world implementation?** Our framework supports both distributed and centralized deployments: the distributed approach employs a federated learning architecture with central training and edge-based parameter updates, while the centralized solution addresses scalability through high-performance servers and algorithmic optimizations like DDIM (Song et al., 2020) acceleration. For comprehensive coverage of intersection failures (Kriging) and sensor-specific issues (Random missing), we recommend a hybrid approach, implementing centralized data missing recovery algorithms alongside edge-based noise reduction using cost-effective hardware for accelerated denoising. This balanced strategy ensures robustness, real-time performance, and attack resilience across all deployment scenarios.

## 6. Related Works

**TSC Algorithms.** Since the introduction of static fixed-time plans in 1958 (Webster, 1958), TSC systems have evolved significantly. Systems like SCOOT (Hunt et al., 1982) and SCATS (Lowrie, 1990) rely on expert-designed plans with predefined thresholds, lacking dynamic adaptability to changing traffic conditions. The advent of RL methods has marked a paradigm shift, leveraging real-time traffic data to optimize signal management through trial-and-error, outperforming traditional approaches. RL models

in TSC vary from value-based (Abdulhai et al., 2003; Wei et al., 2018), policy-based (Mousavi et al., 2017), to actor-critic frameworks (Aslani et al., 2018; Wu et al., 2022), with state and reward designs incorporating features like queue length (Varaiya, 2013; Wu et al., 2021; Li et al., 2025), vehicle counts (Wei et al., 2019a; Xu et al., 2021), or travel time (Zheng et al., 2019). Advanced-Colight method (Zhang et al., 2022a) stands out, achieving state-of-the-art results by using running and waiting vehicles to model lane capacity relationships.

**Robust RL.** In RL, disturbances cause errors, leading to Robust RL for improved reliability, split into "training-time" and "testing-time" robustness. "Training-time" adds noise during training for adaptability (Zhang et al., 2022c; Ye et al., 2023). "Testing-time" trains in clean settings and tests under disruptions (Yang et al., 2022; Panaganti et al., 2022). Our RobustLight focuses on "testing-time" robustness for real-world attack resilience. "Testing-time" robustness covers state, action, and transition/reward perturbations. State perturbation uses neural networks and SA-MDP (Zhang et al., 2021; 2020) or conservative actions (Yang et al., 2022). Action perturbation employs adversarial training (Tan et al., 2020) or optimal policies (Liu et al., 2023). Transition/reward perturbations use Markov games (Pinto et al., 2017; Gleave et al., 2019) or robust Bellman operators (Panaganti et al., 2022). We prioritize "testing-time" robustness for real-world attack resilience.

**Diffusion Model.** Diffusion models, first used for image generation (Ho et al., 2020b), excel in RL control tasks. Diffusion RL splits into online and offline settings. Online, Yang et al. (Yang et al., 2023) applied diffusion as a policy with model-free control. Offline, Ajay et al. (Ajay et al., 2022) generated trajectories for decisions, and Wang et al. (Wang et al., 2022) introduced Diffusion-QL, merging TD3+BC (Fujimoto & Gu, 2021) with behavior cloning. Online TSC algorithms (Wei et al., 2019a; Chen et al., 2020; Zhang et al., 2022a) falter under state attacks and missing data, risking congestion. RobustLight, our proposed algorithm, enhances online TSC resilience by recovering data without altering existing systems, offering a novel solution for research and industry.

## 7. Conclusion.

In this paper, we introduced RobustLight, designed to address abnormal TSC tasks. By leveraging the denoising properties of the diffusion model, RobustLight effectively handles noise interference and incomplete state information in real-world environments. Empirical results demonstrate its robustness and effectiveness, significantly enhancing the security of RL-based TSC strategies and strengthening traditional TSC algorithms, thereby improving the safety and integrity of TSC systems.

## Acknowledgements

We sincerely thank our collaborators for their invaluable feedback and meticulous proofreading of this manuscript. This work was supported in part by the National Key R&D Program of China (Grant No. 2024YFB2906503) and the National Natural Science Foundation of China (Grant No. 62032002).

## Impact Statement

This paper presents work whose goal is to advance the field of Machine Learning. There are many potential societal consequences of our work, none which we feel must be specifically highlighted here.

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

# A. Appendix: Settings.

## A.1. Unet Network Structure

The unet structure is shown in Tables 7, 8, and 9.

*Table 7.* TemporalUnet Structure

| Layer | Layer (input size, output size) |
|---|---|
| **state_encoder** | Linear(state_dim, hidden_size)
Mish()
Linear(hidden_size, state_dim) |
| **action_encoder** | Linear(action_dim, hidden_size)
Mish()
Linear(hidden_size, action_dim) |
| **time_mlp** | SinusoidalPosEmb(hidden_size,hidden_size)
Linear(hidden_size, hidden_size*2)
Mish()
Linear(hidden_size*2, hidden_size) |
| **downs** | ResidualTemporalBlock(hidden_size,hidden_size*2)
ResidualTemporalBlock(hidden_size*2,hidden_size*2)
Identity()
Downsample1d(hidden_size*2,hidden_size*2)
ResidualTemporalBlock(hidden_size*2,hidden_size*4)
ResidualTemporalBlock(hidden_size*4,hidden_size*2)
Identity()
Identity() |
| **ups** | ResidualTemporalBlock(hidden_size*8,hidden_size*2)
ResidualTemporalBlock(hidden_size*2,hidden_size*2)
Identity()
ConvTranspose1d(hidden_size*2,hidden_size*2) |
| **mid_block1** | ResidualTemporalBlock(hidden_size*4,hidden_size*4)
ResidualTemporalBlock |
| **mid_attn** | Identity() |
| **mid_block2** | ResidualTemporalBlock(hidden_size*4,hidden_size*4) |
| **final_conv** | Conv1dBlock(hidden_size*2,hidden_size*2)
Conv1d(hidden_size*2,hidden_size/4) |
| **mid_layer** | Linear(out_horizon*hidden_size/4+
(hidden_size*3)/2+hidden_size,hidden_size*2)
Mish()
Linear(hidden_size*2, hidden_size*2)
Mish()
Linear(hidden_size*2, hidden_size*2)
Mish() |
| **final_layer** | Linear(hidden_size*2, hidden_size/2) |

*Table 8.* ResidualTemporal Block Structure

| Layer | | Details |
|---|---|---|
| **ResidualTemporalBlock** | **blocks** | Conv1dBlock(input_dim,output_dim) |
| | | Conv1dBlock(output_dim,output_dim) |
| | **time_mlp** | Mish() |
| | | Linear(output_dim, output_dim*2) |
| | | Rearrange('batch t - batch t 1') |
| | **residual_conv** | Conv1d(input_dim, output_dim) |

*Table 9.* Conv1dBlock Structure

| Layer | Details |
|---|---|
| **Conv1dBlock** | Conv1d(input_dim,output_dim,Kernel Size, Stride, Padding) |
| | Rearrange('batch channels horizon' → 'batch channels 1 horizon' |
| | 'batch channels 1 horizon' → 'batch channels horizon') |
| | GroupNorm(output_dim, Group, eps, Affine: True) |
| | Mish() |

## A.2. Hyperparameter

By effectively tuning these hyperparameters, users can optimize RobustLight performance for their specific data recovery tasks, achieving better accuracy and robustness in handling missing or corrupted data. The detailed settings are summarized in Table 10.

*Table 10.* Hyperparameters

| Hyperparameter type | Diffusion policy Hyperparameter | Setting |
|---|---|---|
| UNet hyperparameter | embed_dim | 64 |
| | state_dim | 12/24 |
| | action_dim | 4 |
| Diffusion training hyperparameter | non_markovian_step | 6 |
| | condition_length | 4 |
| | beta schedule $a, b, c$ | 2.1190,25.06,-2.5446 |
| | discount($\gamma$) | 0.99 |
| | target critic($\tau$) | 0.005 |
| | diffusion timstep | 100 |
| | batch size | 64 |
| | buffer capacity | 12000 |
| | optimizer | Adam |
| | learning rate | 0.0003 |
| | epochs | 50 |
| | action gradient steps | 20 |
| TSC RL agent training hyperparameter | discount($\gamma$) | 0.8 |
| | target critic($\tau$) | 0.95 |
| | buffer capacity | 12000 |
| | epochs | 100 |
| | batch_size | 20 |
| | learning_rate | 0.001 |
| | target update time | 5 |
| | normal factor | 20 |
| | loss function | MSE |
| | optimizer | Adam |

*Table 11.* Performance in terms of ATT. %25 means the $sensor_N$, %50 means the $sensor_N$, $sensor_E$ missing data.

| Dataset | Mask Scale | FixedTime base | MaxPressure base | MaxPressure RobustLight | Advanced-MaxPressure base | Advanced-MaxPressure RobustLight | Advanced-MpLight base | Advanced-MpLight RobustLight | CoLight base | CoLight RobustLight | Advanced-CoLight base | Advanced-CoLight RobustLight |
|---|---|---|---|---|---|---|---|---|---|---|---|---|
| JiNan1 | 25% | 428.11±0.00 | 286.75±0.00 | **279.52±1.07** | 265.36±0.0 | **263.07±1.09** | 524.48±199.99 | **260.36±4.47** | 316.76±15.24 | **276.02±3.44** | 253.58±1.84 | **252.88±1.3** |
|  | 50% |  | 308.17±0.0 | **299.56±2.88** | 277.53±0.0 | **275.13±1.83** | 278.37±2.47 | **270.44±5.09** | 312.07±4.91 | 321.85±5.63 | 313.69±12.85 | **304.13±11.42** |
| JiNan2 | 25% | 368.76±0.00 | 252.12±0.00 | 283.78±10.61 | 253.71±0.0 | **250.77±0.85** | 268.98±34.52 | **242.73±0.47** | 267.43±2.42 | **260.0±3.18** | 237.03±1.09 | 243.49±1.86 |
|  | 50% |  | 276.93±0.0 | **272.67±3.52** | 269.6±0.0 | **266.67±1.58** | 258.02±1.38 | **242.01±25.74** | 279.01±4.46 | **271.44±2.54** | 267.85±6.47 | 282.31±7.6 |
| JiNan3 | 25% | 383.01±0.00 | 251.22±0.00 | **250.09±0.83** | 248.68±0.0 | **245.66±0.74** | 259.62±40.43 | **233.4±3.43** | 267.17±3.04 | **253.9±1.16** | 242.36±1.7 | **239.41±1.7** |
|  | 50% |  | 273.32±0.0 | **270.64±4.39** | 265.72±0.0 | **263.77±1.53** | 255.55±1.54 | 343.01±52.36 | 288.54±5.05 | **273.83±2.21** | 304.03±26.17 | **268.88±9.01** |
| HangZhou1 | 25% | 495.57±0.00 | 306.21±0.00 | **307.8±1.58** | 397.12±0.0 | **301.53±4.38** | 505.87±94.27 | **305.12±7.49** | 376.22±19.3 | **320.05±6.62** | 303.94±10.84 | **290.02±7.84** |
|  | 50% |  | 392.9±0.0 | 524.5±48.28 | 528.11±0.0 | **433.36±38.17** | 428.21±42.96 | **346.04±10.38** | 467.12±12.2 | **362.18±9.7** | 511.01±56.7 | **375.96±47.84** |
| HangZhou2 | 25% | 406.65±0.00 | 352.17±0.00 | **349.81±1.19** | 378.94±0.0 | **330.82±3.91** | 382.15±10.52 | **349.4±2.56** | 383.97±15.87 | **362.11±8.16** | 347.53±18.69 | **334.01±7.22** |
|  | 50% |  | 425.14±0.0 | **405.95±6.37** | 452.73±0.0 | **422.18±12.57** | 436.27±18.12 | **413.07±13.63** | 434.66±7.18 | **415.51±7.82** | 444.82±41.36 | **383.46±17.09** |

*Table 12.* Performance of in terms of ATT. %25 means the $sensor_E$, %50 means the $sensor_E$, $sensor_S$ missing data.

| Dataset | Mask Scale | FixedTime base | MaxPressure base | MaxPressure RobustLight | Advanced-MaxPressure base | Advanced-MaxPressure RobustLight | Advanced-MpLight base | Advanced-MpLight RobustLight | CoLight base | CoLight RobustLight | Advanced-CoLight base | Advanced-CoLight RobustLight |
|---|---|---|---|---|---|---|---|---|---|---|---|---|
| JiNan1 | 25% | 428.11±0.00 | 308.68±0.0 | **292.73±4.19** | 288.06±0.0 | **276.14±2.23** | 535.51±110.15 | **360.76±74.58** | 353.3±4.7 | **328.99±2.28** | 291.08±4.77 | 298.17±8.72 |
|  | 50% |  | 704.67±0.0 | 877.64±81.64 | 1138.54±0.0 | **837.9±100.32** | 1168.06±45.85 | **838.9±76.63** | 967.5±21.19 | **955.55±1.28** | 635.65±21.05 | **615.93±31.48** |
| JiNan2 | 25% | 368.76±0.00 | 265.92±0.0 | **259.68±6.4** | 273.48±0.0 | **257.17±1.6** | 817.22±231.93 | **275.46±46.36** | 356.79±4.03 | **281.06±2.07** | 264.35±5.53 | 254.08±4.13 |
|  | 50% |  | 758.94±0.0 | **641.38±38.7** | 1106.91±0.0 | **959.47±69.72** | 1178.75±106.23 | **809.77±64.74** | 1125.51±37.74 | **1044.55±16.55** | 659.29±39.5 | **650.12±39.04** |
| JiNan3 | 25% | 383.01±0.00 | 261.32±0.0 | **260.4±1.35** | 264.86±0.0 | **252.11±1.38** | 545.89±179.58 | **325.41±44.77** | 305.41±7.81 | **271.38±3.22** | 260.69±2.7 | **253.03±4.76** |
|  | 50% |  | 668.31±0.0 | 701.76±53.94 | 1254.97±0.0 | **727.59±91.6** | 1114.57±64.26 | **761.78±143.2** | 1022.03±23.29 | **801.03±88.64** | 643.5±45.69 | **596.76±42.72** |
| HangZhou1 | 25% | 495.57±0.00 | 365.3±0.0 | 426.3±51.56 | 450.99±0.0 | **300.44±2.19** | 619.52±133.36 | **410.38±60.16** | 432.58±6.51 | **335.26±10.09** | 381.1±62.66 | **340.48±22.93** |
|  | 50% |  | 636.51±0.0 | 754.16±33.52 | 1005.01±0.0 | **781.38±69.53** | 825.5±49.93 | **424.8±55.41** | 912.27±43.76 | **462.48±37.71** | 602.16±109.41 | **516.18±19.13** |
| HangZhou2 | 25% | 406.65±0.00 | 381.51±0.0 | **371.48±6.28** | 414.65±0.0 | **351.19±4.67** | 396.42±23.46 | **353.43±2.89** | 426.9±2.24 | **379.94±7.1** | 377.86±28.22 | **352.87±10.5** |
|  | 50% |  | 554.02±0.0 | 552.24±23.82 | 718.93±0.0 | **534.73±38.6** | 687.16±62.93 | **447.47±17.12** | 693.78±34.59 | **471.14±12.57** | 501.53±48.97 | **446.97±11.96** |

*Table 13.* Performance in terms of ATT. %25 means the $sensor_S$, %50 means the $sensor_W$, $sensor_S$ missing data.

| Dataset | Mask Scale | FixedTime base | MaxPressure base | MaxPressure RobustLight | Advanced-MaxPressure base | Advanced-MaxPressure RobustLight | Advanced-MpLight base | Advanced-MpLight RobustLight | CoLight base | CoLight RobustLight | Advanced-CoLight base | Advanced-CoLight RobustLight |
|---|---|---|---|---|---|---|---|---|---|---|---|---|
| JiNan1 | 25% | 428.11±0.00 | 307.99±0.02 | 318.51±9.13 | 294.96±0.0 | **272.62±1.45** | 314.89±29.39 | 456.42±158.17 | 563.6±108.06 | **347.75±13.07** | 277.76±4.93 | 297.89±6.94 |
|  | 50% |  | 422.76±0.0 | 537.52±16.47 | 320.54±0.0 | **319.24±5.6** | 326.15±6.18 | 327.87±5.09 | 783.84±298.76 | **377.02±25.84** | 379.57±13.3 | 431.91±25.96 |
| JiNan2 | 25% | 368.76±0.00 | 263.82±0.0 | **258.16±2.01** | 268.66±0.0 | **256.44±1.63** | 266.23±20.64 | **249.71±1.88** | 286.31±5.87 | **266.66±2.43** | 258.47±2.06 | **257.17±3.87** |
|  | 50% |  | 306.72±0.0 | 459.55±9.48 | 294.71±0.0 | **288.55±4.1** | 293.4±18.01 | 304.9±2.08 | 342.81±36.26 | **295.67±6.86** | 307.66±14.51 | 377.25±28.18 |
| JiNan3 | 25% | 383.01±0.00 | 270.41±0.0 | 275.75±7.51 | 262.33±0.0 | **254.85±1.56** | 453.95±92.19 | **250.51±1.99** | 315.87±7.89 | **273.27±5.88** | 277.3±28.08 | **261.9±5.56** |
|  | 50% |  | 323.21±0.0 | 459.55±9.48 | 306.58±0.0 | **286.77±4.06** | 330.86±22.73 | 337.5±44.54 | 320.0±2.34 | 331.79±18.83 | 341.43±9.31 | 365.48±16.22 |
| HangZhou1 | 25% | 495.57±0.00 | 313.71±0.0 | 319.2±11.11 | 445.59±0.0 | **364.62±14.37** | 416.28±22.38 | **391.16±55.98** | 434.97±1.84 | **396.72±25.87** | 408.48±21.16 | **345.09±11.53** |
|  | 50% |  | 430.52±0.0 | 569.06±7.55 | 632.61±0.0 | **484.92±19.65** | 460.54±12.55 | **388.82±11.75** | 577.86±20.19 | **511.12±23.68** | 656.46±29.04 | **514.43±46.55** |
| HangZhou2 | 25% | 406.65±0.00 | 361.69,0.0 | 364.5±2.2 | 394.35±0.0 | **369.94±4.28** | 398.64±5.08 | **379.31±8.41** | 406.21±4.92 | **396.05±8.35** | 393.66±12.79 | **361.81±7.71** |
|  | 50% |  | 387.58±0.0 | 461.11±4.51 | 446.4±0.0 | **411.78±3.03** | 458.86±16.85 | 467.39±0.68 | 437.73±16.48 | **411.47±13.55** | 462.53±17.42 | 419.92±9.52 |

*Table 14.* Performance of ATT comparison between the native diffusion model, LSTM model and our improved diffusion model.

| Dataset | Noise Type | Noise Scale | CoLight base | CoLight RobustLight | CoLight Diffusion Linear-Beta | CoLight Diffusion Cosine-Beta | CoLight Diffusion vp-Beta | CoLight LSTM | Advanced-CoLight base | Advanced-CoLight RobustLight | Advanced-CoLight Diffusion Linear-Beta | Advanced-CoLight Diffusion Cosine-Beta | Advanced-CoLight Diffusion vp-Beta | Advanced-CoLight LSTM |
|---|---|---|---|---|---|---|---|---|---|---|---|---|---|---|
| JiNan1 | Gaussian | 3.5 | 287.76±1.52 | **282.35±1.96** | 287.08±2.23 | 286.89±1.17 | 288.6±1.60 | 1219.5±67.02 | 320.57±2.82 | **294.57±2.02** | 324.04±3.04 | 323.65±8.30 | 324.3±6.66 | 892.0±213.7 |
|  |  | 4.0 | 314.93±28.28 | **288.07±2.27** | 294.76±1.35 | 1289.13±0.58 | 296.62±2.38 | 1225.65±59.28 | 338.93±15.32 | **298.23±5.16** | 326.83±6.56 | 1230.28±6.59 | 342.47±8.47 | 914.18±215.7 |
|  | U-rand | 3.5 | 407.41±24.57 | **312.27±4.07** | 407.38±22.23 | 424.39±14.73 | 405.22±23.74 | 1102.77±134.88 | 449.13±21.14 | **358.22±7.02** | 433.95±9.03 | 437.24±12.43 | 445.68±22.08 | 786.75±153.39 |
|  |  | 4.0 | 352.63±7.85 | **320.48±3.29** | 431.15±24.95 | 449.54±7.18 | 426.82±26.03 | 1134.31±119.32 | 460.46±22.41 | **360.82±4.79** | 456.95±6.03 | 453.61±18.91 | 458.69±21.52 | 794.28±152.55 |
|  | MAD | 3.5 | 487.69±50.75 | **277.37±1.16** | 471.97±33.31 | 630.83±37.37 | 743.38±1.03 | 1302.95±1.19 | 479.63±8.82 | **283.13±1.56** | 370.0±9.50 | 356.21±5.30 | 360.58±6.34 | 968.35±150.25 |
|  |  | 4.0 | 555.71±64.93 | **279.15±1.22** | 622.46±13.46 | 778.2±14.04 | 810.48±1.34 | 1303.05±1.17 | 520.09±7.12 | **288.07±2.27** | 383.12±3.28 | 366.46±6.34 | 372.44±5.32 | 983.57±167.12 |
|  | MinQ | 3.5 | 683.38±94.91 | **277.93±3.47** | 713.04±19.43 | 485.79±18.87 | 497.0±4.35 | 1296.72±16.26 | 394.39±50.27 | **323.25±20.54** | 508.66±21.35 | 514.2±30.03 | 523.33±17.34 | 1095.13±163.77 |
|  |  | 4.0 | 716.58±134.92 | **281.13±1.96** | 811.2±32.13 | 529.54±32.45 | 553.46±2.36 | 1296.72±16.26 | 394.63±47.37 | **331.52±10.64** | 532.77±12.30 | 570.18±14.03 | 576.11±15.04 | 1091.69±158.39 |
|  | Sensor | 25% | 400.93±19.66 | **372.93±7.43** | 409.63±3.37 | 1293.77±5.93 | 538.01±183.96 | 1290.52±110.71 | 326.14±16.52 | **298.35±16.32** | 350.96±17.32 | 1291.53±23.83 | 402.87±33.58 | 1035.39±156.69 |
|  | Damage | 50% | 849.61±77.93 | **766.91±81.83** | 771.24±22.29 | 1295.14±23.31 | 789.67±132.54 | 1297.65±112.3 | 732.37±52.93 | **680.37±97.32** | 689.13±98.43 | 1295.74±34.86 | 760.2±36.66 | 1088.77±160.11 |
| HangZhou1 | Gaussian | 3.5 | 356.33±6.38 | **322.62±5.96** | 362.66±5.36 | 363.8±0.92 | 374.12±9.71 | 435.97±103.99 | 480.38±22.93 | **327.98±2.45** | 480.53±33.55 | 482.46±24.13 | 504.29±30.48 | 1123.4±341.28 |
|  |  | 4.0 | 371.97±12.44 | **337.58±5.68** | 378.45±5.2 | 1121.71±2.74 | 408.76±18.96 | 441.03±122.33 | 510.38±25.17 | **331.21±3.65** | 510.16±22.34 | 1115.67±3.63 | 546.12±27.14 | 1140.26±325.02 |
|  | U-rand | 3.5 | 647.64±54.89 | **435.33±20.02** | 660.63±52.6 | 643.53±46.33 | 654.66±49.97 | 628.22±124.08 | 717.12±70.08 | **473.85±32.68** | 720.43±68.66 | 724.03±67.08 | 720.34±61.62 | 991.97±312.59 |
|  |  | 4.0 | 475.49±29.78 | **441.33±17.05** | 704.63±44.9 | 674.75±47.63 | 690.65±52.65 | 656.45±129.92 | 738.51±63.18 | **487.74±31.44** | 742.7±64.37 | 751.11±62.34 | 756.38±65.69 | 1025.41±324.38 |
|  | MAD | 3.5 | 402.67±17.42 | **324.55±6.63** | 414.88±3.63 | 530.83±13.64 | 522.02±6.34 | 724.26±307.81 | 463.64±33.44 | **309.24±0.94** | 479.13±10.69 | 499.48±1.03 | 477.33±9.91 | 934.45±64.37 |
|  |  | 4.0 | 419.07±18.26 | **331.47±8.62** | 429.54±9.31 | 570.98±4.63 | 529.69±9.31 | 782.48±292.96 | 491.64±39.78 | **352.58±3.71** | 510.51±6.25 | 518.84±4.32 | 515.24±33.63 | 1124.45±32.27 |
|  | MinQ | 3.5 | 466.14±70.37 | **317.36±3.96** | 538.68±4.03 | 402.27±1.36 | 402.33±5.85 | 762.59±267.59 | 449.54±8.22 | **406.32±6.41** | 488.99±16.95 | 495.9±8.34 | 487.39±9.93 | 783.46±28.32 |
|  |  | 4.0 | 460.29±62.45 | **331.93±3.91** | 570.33±5.34 | 407.07±6.04 | 405.55±6.24 | 763.68±273.87 | 482.98±7.29 | **430.53±8.49** | 517.86±12.74 | 524.97±9.14 | 535.21±11.71 | 928.32±123.32 |
|  | Sensor | 25% | 490.36±13.84 | **424.36±13.22** | 490.01±1.54 | 1123.98±10.6 | 434.56±27.88 | 602.42±111.57 | 401.87±48.31 | **328.26±12.67** | 386.13±29.28 | 1122.75±3.18 | 400.57±49.43 | 1581.35±73.21 |
|  | Damage | 50% | 786.32±40.86 | **542.91±108.24** | 776.83±25.33 | 1121.73±12.75 | 560.45±131.35 | 734.51±70.43 | 842.29±235.36 | **541.56±122.96** | 551.44±40.16 | 1123.49±3.03 | 611.57±121.67 | 1638.32±14.1 |

*Table 15.* **Performance of ATT in JiNan, HangZhou. "-" implies that traditional algorithms are not adapted to MinQ and MAD attacks. Our RobustLight recovers the state of traditional and RL-based TSC algorithms to evaluate the performance.**

| Dataset | Noise Type | Noise Scale | FixedTime | MaxPressure | | Advanced-MaxPressure | | Advanced-MpLight | | CoLight | | Advanced-CoLight | |
|---|---|---|---|---|---|---|---|---|---|---|---|---|---|
| | | | base | base | RobustLight | base | RobustLight | base | RobustLight | base | RobustLight | base | RobustLight |
| $JiNan_2$ | Gaussian | 3.5 | 368.76±0.00 | 279.05±1.56 | **276.14±1.49** | 275.95±1.22 | **274.67±1.58** | 383.41±58.34 | **276.02±1.72** | 295.87±17.16 | **265.21±2.02** | 326.04±10.78 | **289.84±4.68** |
| | | 4.0 | | 282.99±1.22 | **279.89±1.41** | 279.63±1.15 | **276.56±1.74** | 455.56±110.18 | **281.88±2.91** | 336.38±61.85 | **271.65±1.57** | 340.84±17.41 | **298.63±5.04** |
| | U-rand | 3.5 | | 301.51±2.82 | **290.84±2.46** | 299.75±1.16 | **297.34±1.39** | 548.04±98.46 | **323.93±11.34** | 653.03±150.29 | **380.63±68.09** | 717.76±169.73 | **361.72±28.89** |
| | | 4.0 | | 306.29±3.09 | **294.39±3.25** | 304.75±1.09 | **302.97±1.91** | 514.01±116.27 | **327.93±11.35** | 359.26±12.55 | **354.02±53.63** | 741.76±181.97 | **361.66±30.15** |
| | MAD | 3.5 | | - | - | - | - | 285.67±11.23 | **263.21±1.17** | 565.93±39.44 | **262.05±1.48** | 325.87±8.34 | **271.58±1.52** |
| | | 4.0 | | - | - | - | - | 297.97±18.32 | **268.94±1.13** | 588.27±47.11 | **263.45±1.65** | 337.42±8.64 | **271.65±1.57** |
| | MinQ | 3.5 | | - | - | - | - | 283.37±3.09 | **264.48±1.83** | 516.49±102.73 | **261.41±1.34** | 323.07±7.45 | **300.31±5.44** |
| | | 4.0 | | - | - | - | - | 289.43±3.78 | **270.98±1.83** | 541.17±112.48 | **268.17±2.29** | 346.51±3.85 | **314.29±3.67** |
| $HangZhou_2$ | Gaussian | 3.5 | 406.65±0.00 | 360.56±2.45 | **358.57±1.71** | 345.27±1.23 | **342.52±1.23** | 564.65±103.29 | **362.27±3.85** | 349.75±5.13 | **344.63±2.01** | 396.61±9.59 | **368.79±6.27** |
| | | 4.0 | | 363.64±2.77 | **359.93±2.02** | 348.40±1.11 | **346.66±1.02** | 418.41±12.43 | **371.71±3.65** | 352.25±4.43 | **351.58±3.75** | 410.64±10.02 | **371.54±6.94** |
| | U-rand | 3.5 | | 371.21±3.44 | **365.23±2.07** | 362.11±2.13 | **289.62±2.09** | 504.63±9.71 | **437.93±14.49** | 442.07±26.52 | **353.67±3.26** | 472.11±20.71 | **379.27±11.97** |
| | | 4.0 | | 373.89±3.09 | **366.52±2.14** | 366.69±3.53 | **359.17±2.02** | 513.83±11.21 | **445.99±11.53** | 383.27±9.41 | **356.79±4.07** | 487.33±19.17 | **384.24±13.31** |
| | MAD | 3.5 | | - | - | - | - | 345.74±29.73 | **344.52±4.49** | 478.25±15.39 | **349.37±2.54** | 405.37±12.33 | **359.03±9.23** |
| | | 4.0 | | - | - | - | - | 349.48±26.63 | **348.61±5.77** | 490.25±18.66 | **356.53±4.33** | 406.75±5.01 | **359.06±9.23** |
| | MinQ | 3.5 | | - | - | - | - | 347.04±24.07 | **343.38±9.54** | 473.48±11.07 | **347.73±3.85** | 399.87±15.12 | **372.84±4.84** |
| | | 4.0 | | - | - | - | - | 350.15±24.27 | **347.44±9.43** | 484.52±12.88 | **352.13±2.72** | 408.28±8.71 | **385.43±8.06** |

*Table 16.* **Performance Compared with DiffLight.**

| Dataset | Noise Type | Noise Scale | DiffLight | Advanced-Colight-RobustLight |
|---|---|---|---|---|
| $JiNan_1$ | Gaussian | 3.5 | 289.45±3.37 | **294.57±2.02** |
| | U-rand | 3.5 | 311.53±4.17 | **358.22±7.02** |
| | MAD | 3.5 | 384.71±13.25 | **283.13±1.56** |
| | MinQ | 3.5 | 321.98±6.37 | **323.25±20.54** |
| | Mask (Kriging and Random) | 25% | 353.45±34.31 | **320.31±3.21** |
| $HangZhou_1$ | Gaussian | 3.5 | 325.05±2.58 | **327.98±2.45** |
| | U-rand | 3.5 | **365.05±20.67** | 473.85±32.68 |
| | MAD | 3.5 | 366.14±5.34 | **309.24±0.94** |
| | MinQ | 3.5 | 426.14±10.56 | **406.32±6.41** |
| | Mask (Kriging and Random) | 25% | 346.05±41.27 | **296.69±4.67** |

# B. Appendix: Other Experiments

*Table 17.* Recover time comparison.

| Dataset | Noise type | Noise scale | Diffusion step | Advanced Colight Recover time(ms) | Advanced Maxpressure Recover time(ms) |
|---|---|---|---|---|---|
| $JiNan_1$ | Gaussian | 1.0 | 20 | 376.16 | 343.06 |
| | | 0.5 | 20 | 202.98 | 178.95 |
| | U-rand | 1.0 | 16 | 262.89 | 233.45 |
| | | 0.5 | 8 | 142.05 | 124.33 |
| | MAD | 1.0 | 240 | 3792.56 | - |
| | | 0.5 | 120 | 1959.03 | - |
| | MinQ | 1.0 | 240 | 3779.34 | - |
| | | 0.5 | 120 | 1976.44 | - |
| | Sensor damage | 25% | 200 | 2987.34 | 2765.36 |
| | | 50% | 200 | 2997.61 | 2768.24 |
| $HangZhou_2$ | Gaussian | 1.0 | 40 | 417.26 | 447.84 |
| | | 0.5 | 20 | 221.38 | 233.92 |
| | U-rand | 1.0 | 16 | 281.60 | 302.86 |
| | | 0.5 | 8 | 156.32 | 160.16 |
| | MAD | 1.0 | 240 | 5220.45 | - |
| | | 0.5 | 120 | 3581.67 | - |
| | MinQ | 1.0 | 240 | 5348.28 | - |
| | | 0.5 | 120 | 2889.14 | - |
| | Sensor damage | 25% | 200 | 3232.13 | 3552.44 |
| | | 50% | 200 | 3223.55 | 3559.41 |

## C. Appendix: Model Generalization Analysis

In this section, we analyze the model's generalization ability. We transfer the algorithm trained on the $JiNan_1$ dataset to other datasets to observe whether our RobustLight can effectively recover the data under various attacks.

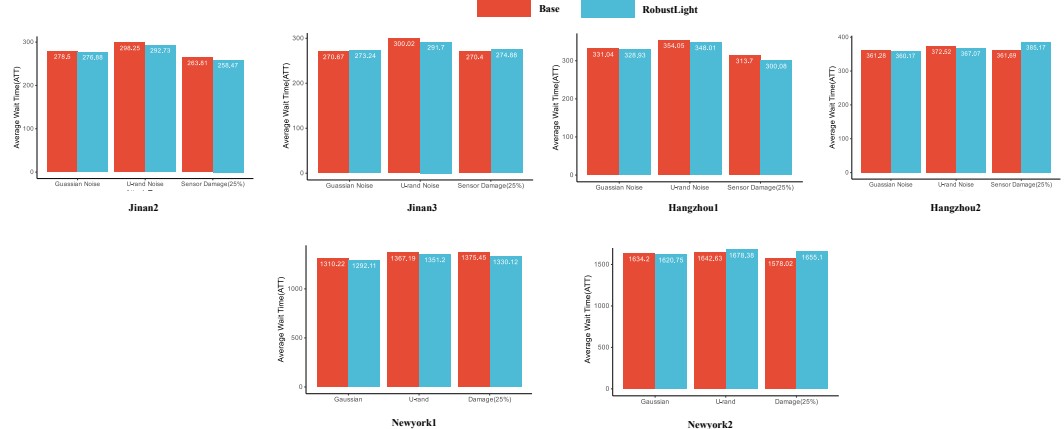

*Figure 7.* **Performance of RobustLight based on Maxpressure transfer by** $JiNan_1$**.**

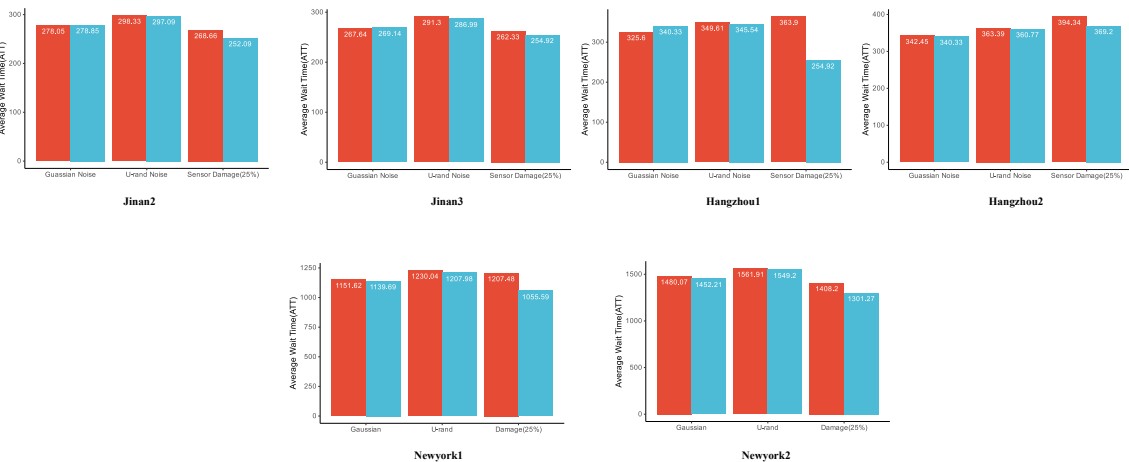

*Figure 8.* **Performance of RobustLight based on Advanced-Maxpressure transfer by** $JiNan_1$**.**

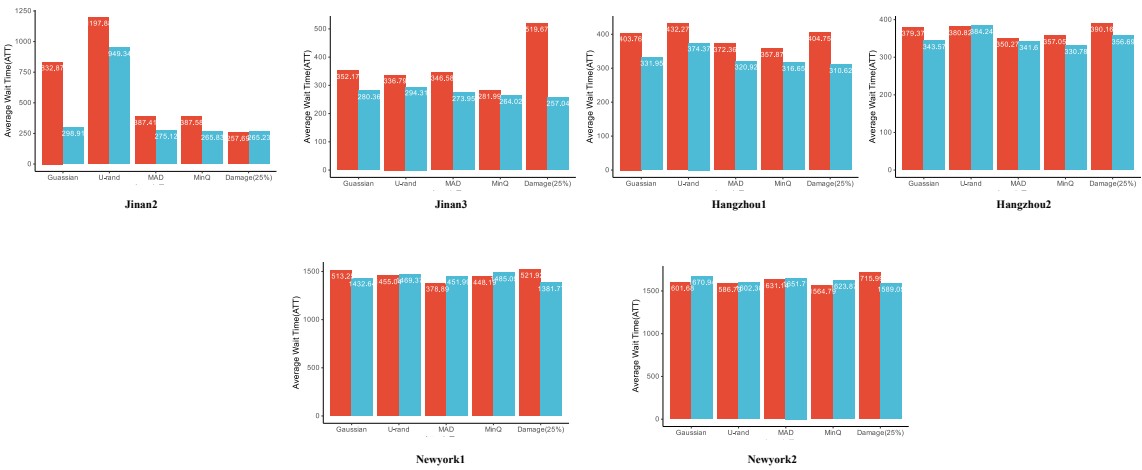

*Figure 9.* **Performance of RobustLight based on Advanced-Mplight transfer by** $JiNan_1$**.**

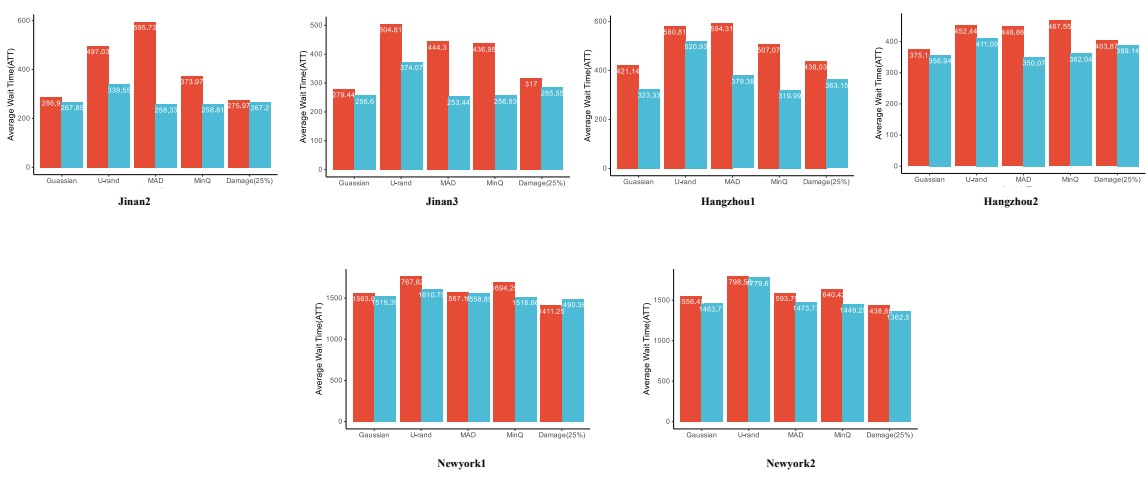

*Figure 10.* **Performance of RobustLight based on Colight transfer by** $JiNan_1$**.**

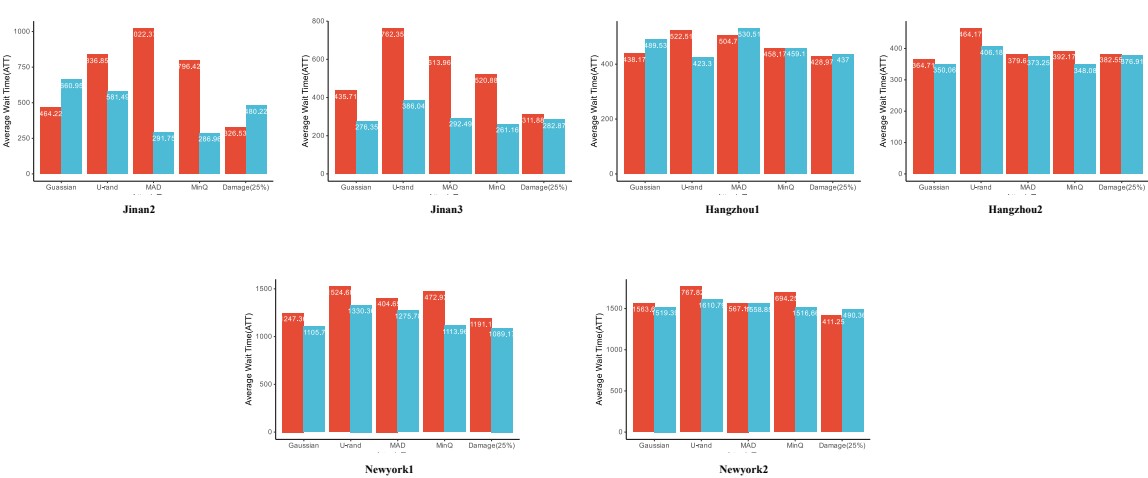

*Figure 11.* **Performance of RobustLight based on Advanced-Colight transfer by** $JiNan_1$**.**

# D. Appendix: Robustness Analysis

In this section, we provide detailed visualizations of the state t-SNE distributions and violin plots after dimensionality reduction under Gaussian noise attacks for different algorithms. As observed, for different algorithms, our RobustLight shows a significantly better recovery of the state distribution in most datasets, making it more closely resemble the original state distribution compared to the attacked state distribution.

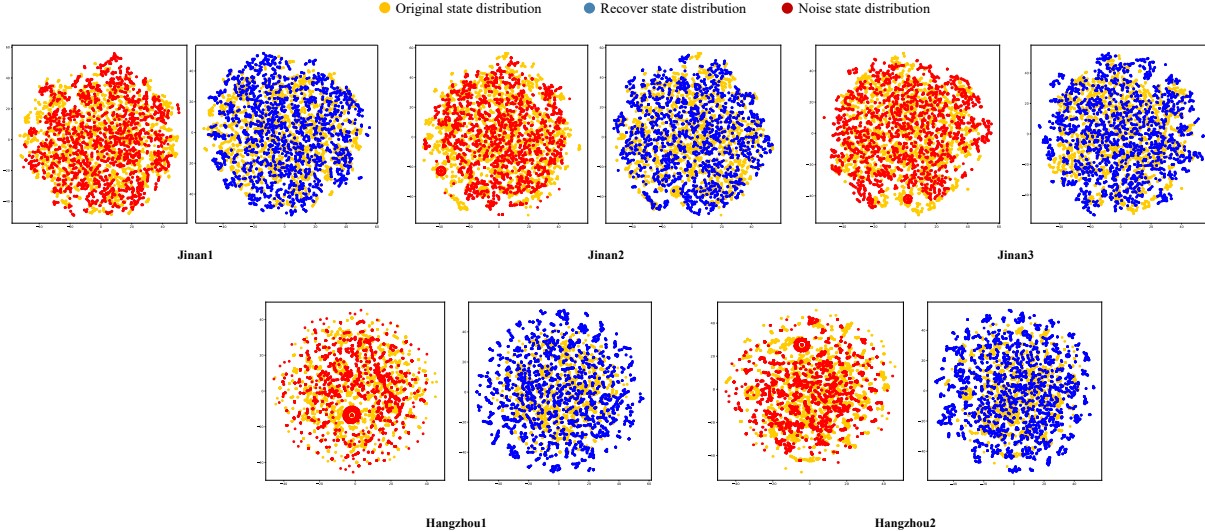

*Figure 12.* **State t-SNE visulization of RobustLight based on Maxpressure in** $HangZhou$ **and** $JiNan$**. Compared to the state distribution after the noise attack, the state distribution is much closer to the original state distribution after using our algorithm. RobustLight has the ability of state recovery.**

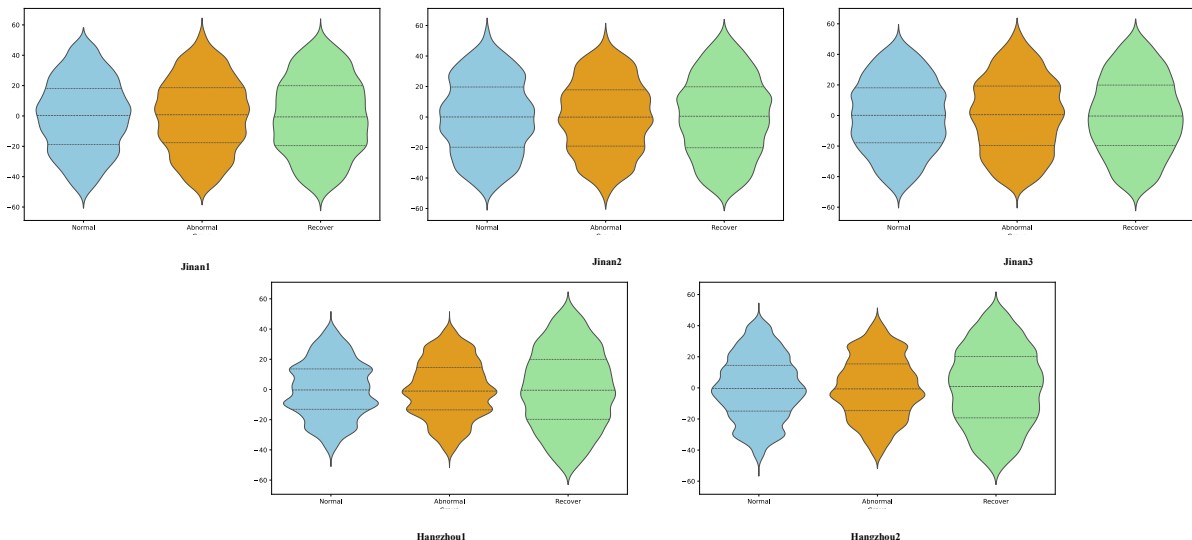

*Figure 13.* **State violin visulization of RobustLight based on Maxpressure in** $HangZhou$ **and** $JiNan$**. Compared to the state distribution after the noise attack, the state distribution is much closer to the original state distribution after using our algorithm. RobustLight has the ability of state recovery.**

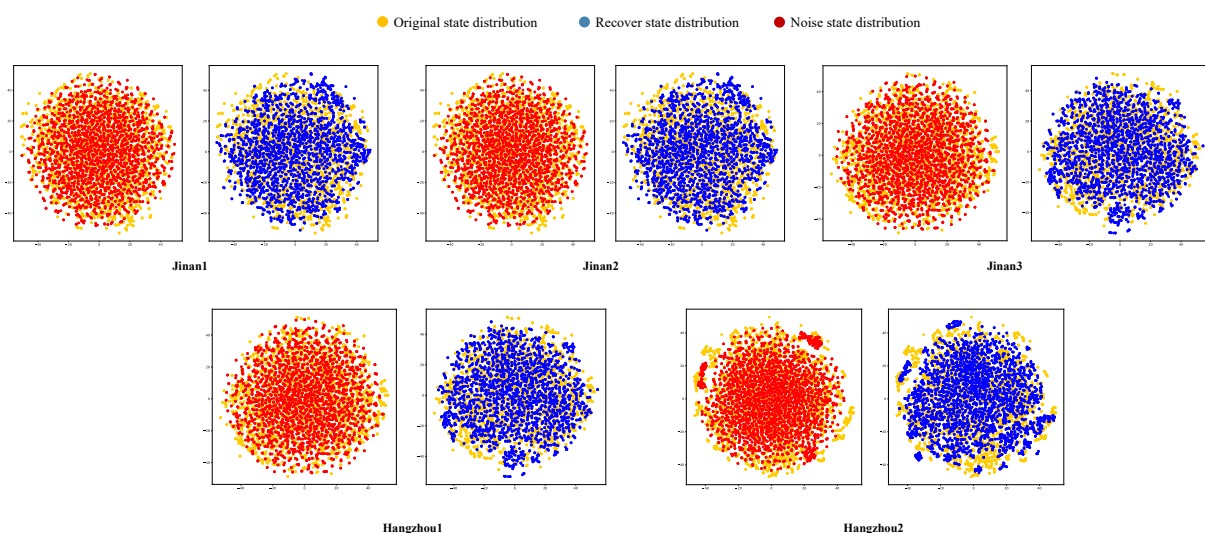

*Figure 14.* **State t-SNE visulization of RobustLight based on AdvancedMaxpressure in** $HangZhou$ **and** $JiNan$**. Compared to the state distribution after the noise attack, the state distribution is much closer to the original state distribution after using our algorithm. RobustLight has the ability of state recovery.**

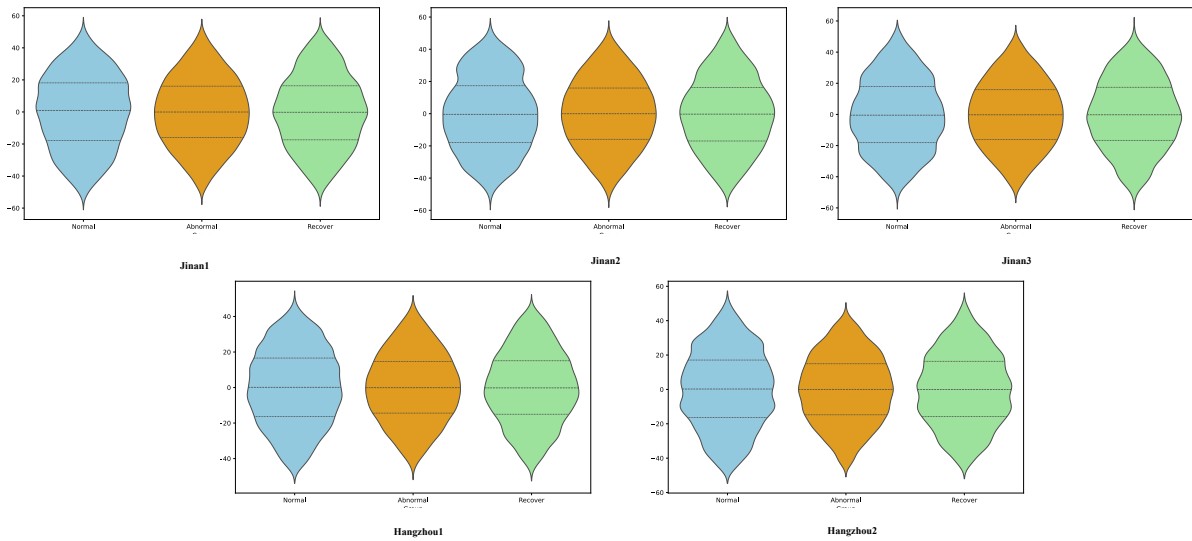

*Figure 15.* **State violin visulization of RobustLight based on AdvancedMaxpressure in** $HangZhou$ **and** $JiNan$**. Compared to the state distribution after the noise attack, the state distribution is much closer to the original state distribution after using our algorithm. RobustLight has the ability of state recovery.**

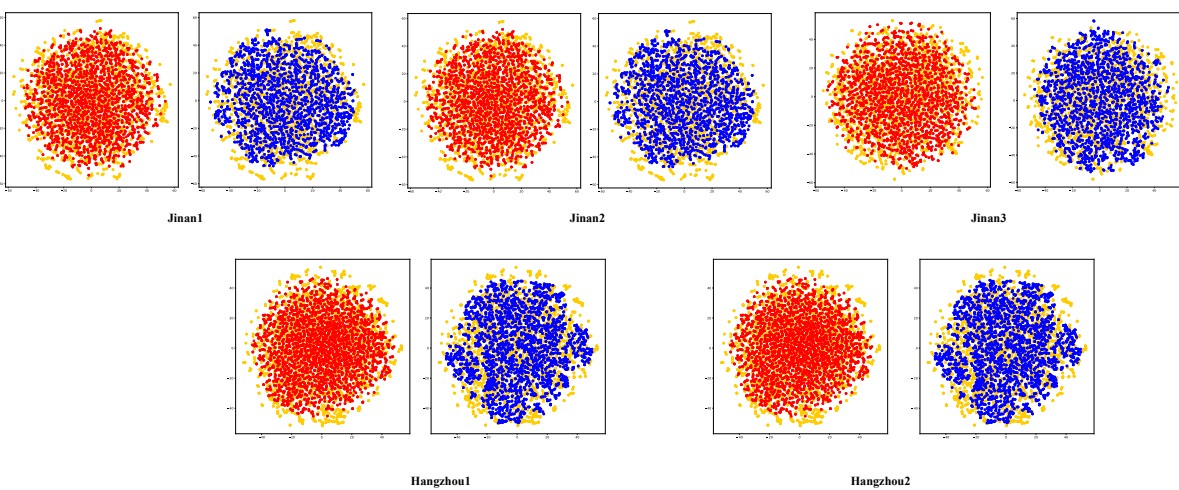

*Figure 16.* **State t-SNE visulization of RobustLight based on Colight in** $HangZhou$ **and** $JiNan$**. Compared to the state distribution after the noise attack, the state distribution is much closer to the original state distribution after using our algorithm. RobustLight has the ability of state recovery.**

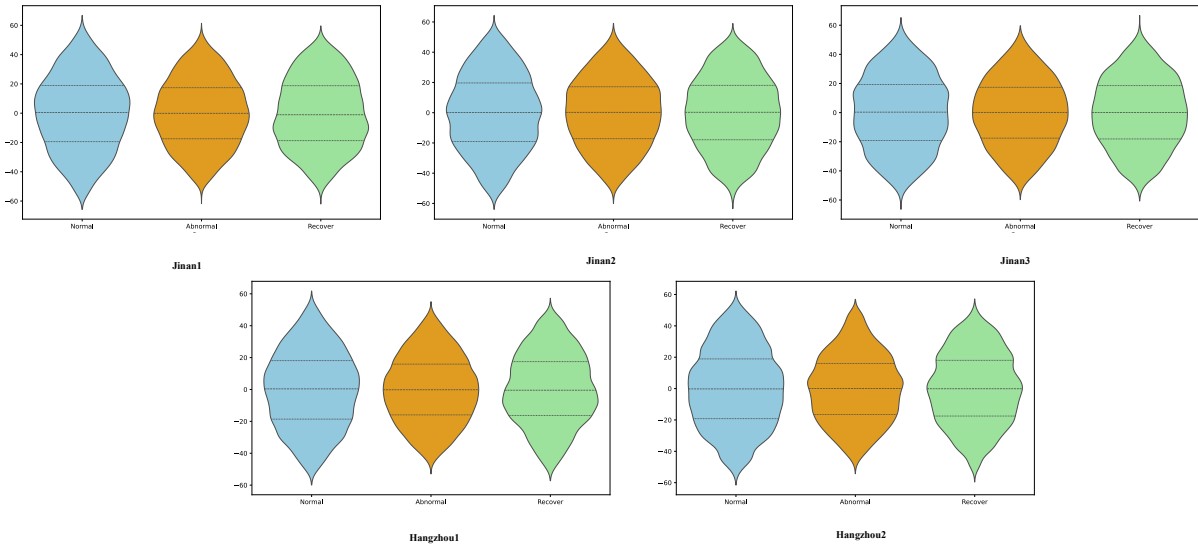

*Figure 17.* **State violin visulization of RobustLight based on Colight in** $HangZhou$ **and** $JiNan$**. Compared to the state distribution after the noise attack, the state distribution is much closer to the original state distribution after using our algorithm. RobustLight has the ability of state recovery.**

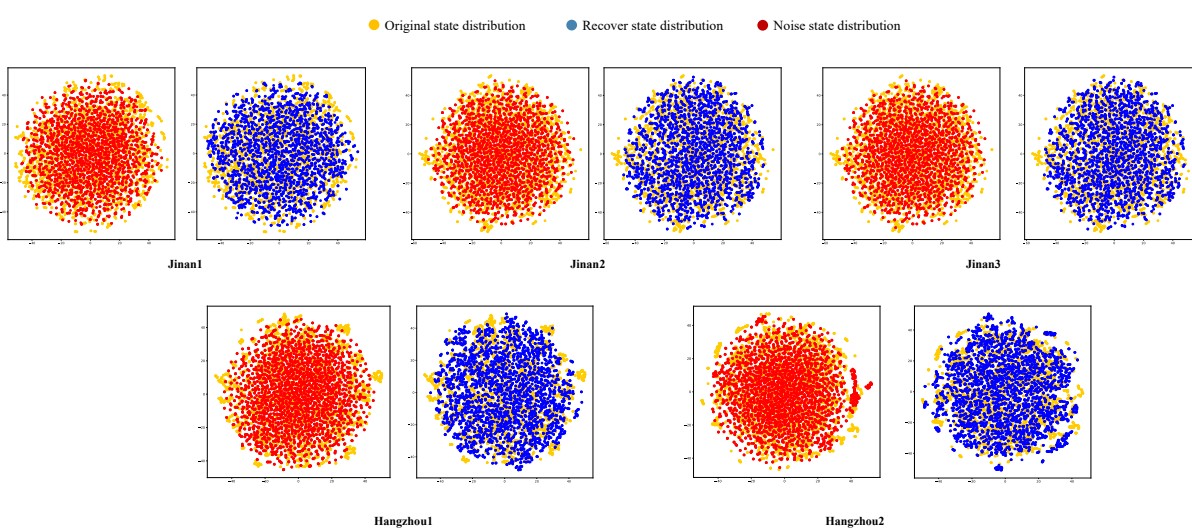

*Figure 18.* **State t-SNE visualization of RobustLight based on AdvancedColight in** $HangZhou$ **and** $JiNan$**. Compared to the state distribution after the noise attack, the state distribution is much closer to the original state distribution after using our algorithm. RobustLight has the ability of state recovery.**

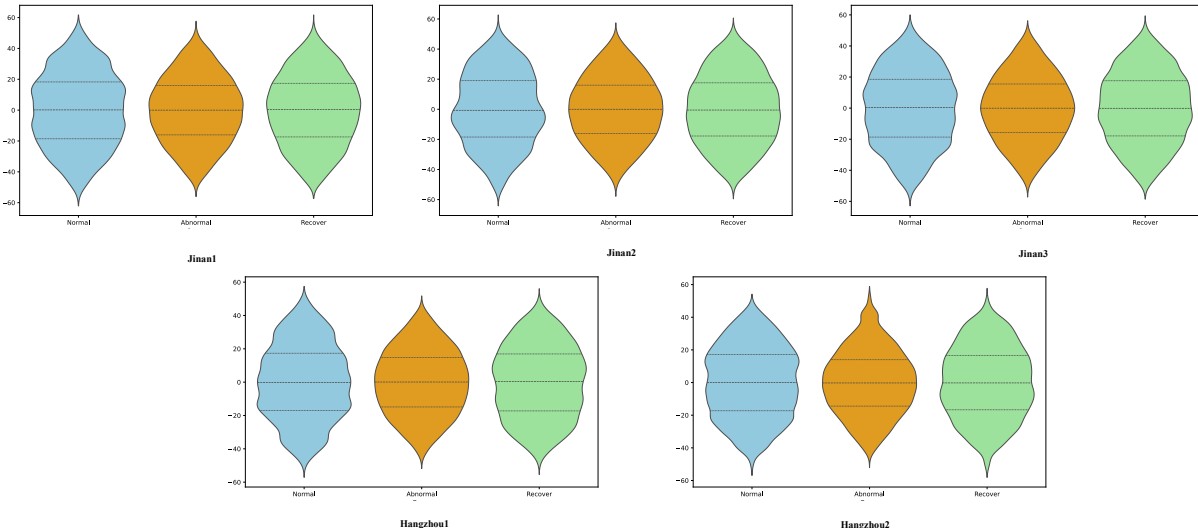

*Figure 19.* **State violin visulization of RobustLight based on AdvancedColight in** $HangZhou$ **and** $JiNan$**. Compared to the state distribution after the noise attack, the state distribution is much closer to the original state distribution after using our algorithm. RobustLight has the ability of state recovery.**

