# OpenReview forum: "RobustLight: Improving Robustness via Diffusion Reinforcement Learning for Traffic Signal Control"
_ICML.cc/2025/Conference — ICML 2025 poster_

### Official Review · Reviewer_Jqz8 · 2025-03-12

**Overall Recommendation:** 3

**Summary:**

The paper introduces ​RobustLight, a novel framework designed to enhance the robustness of Traffic Signal Control (TSC) systems against adversarial attacks and missing data. The authors propose a ​plug-and-play diffusion model​ that integrates with existing TSC platforms to recover from noise attacks and restore missing data in real-time. The framework includes two key algorithms: ​denoise​ and ​repaint, which leverage a ​Dynamic State Infilling (DSI)​ algorithm to train an improved diffusion model online. The authors conduct extensive experiments on real-world datasets, demonstrating that RobustLight significantly improves the performance of TSC systems under various adversarial attacks and missing data scenarios, with up to ​50.43% improvement​ in average travel time compared to systems without RobustLight.

**Claims And Evidence:**

The claims made in the paper are ​well-supported by empirical evidence. The authors provide extensive experimental results on real-world datasets, including JiNan, HangZhou, and New York, to validate the effectiveness of RobustLight. The results show consistent improvements in TSC performance under various adversarial attacks (Gaussian noise, U-rand, MAD, MinQ) and sensor damage scenarios. The authors also compare RobustLight with traditional and RL-based TSC methods, demonstrating its superior robustness and recovery capabilities. The evidence is clear, and the results are statistically significant, with detailed metrics such as ​Average Travel Time (ATT)​ and ​state recovery performance​ provided.

**Essential References Not Discussed:**

There are no essential references not discussed in the paper.

**Experimental Designs Or Analyses:**

The experimental design is ​rigorous and comprehensive. The authors use ​real-world datasets​ from three different cities (JiNan, HangZhou, and New York) to evaluate the performance of RobustLight under various adversarial attacks and sensor damage scenarios. The experiments are well-structured, with clear comparisons between RobustLight and traditional/RL-based TSC methods. The authors also conduct ​ablation studies​ to analyze the impact of different components of RobustLight, providing insights into the contribution of each component to the overall performance. The analysis is thorough, with detailed discussions on the results and their implications for real-world TSC systems.

**Methods And Evaluation Criteria:**

The methods are ​well-designed and innovative, leveraging the strengths of diffusion models and reinforcement learning. The ​Dynamic State Infilling (DSI)​ algorithm is a key contribution, enabling real-time recovery of TSC data. The ​denoise​ and ​repaint​ algorithms are effectively used to handle adversarial attacks and missing data, respectively. The evaluation criteria are appropriate, focusing on ​ATT​ as the primary metric to measure the efficiency of TSC systems. The authors also use ​t-SNE plots​ and ​violin plots​ to visualize the state recovery performance, providing additional insights into the robustness of the proposed framework.

**Other Comments Or Suggestions:**

No.

**Other Strengths And Weaknesses:**

All strengths and weaknesses are mentioned above.

**Questions For Authors:**

- Scalability: The paper mentions that the computational cost increases with the number of intersections. Could the authors elaborate on potential strategies to scale RobustLight for large-scale urban networks with hundreds of intersections?
- Real-World Deployment: While the experiments are conducted on real-world datasets, the paper does not discuss the challenges of deploying RobustLight in real-world TSC systems. What are the practical considerations and potential barriers to real-world implementation?

**Relation To Broader Scientific Literature:**

The key contributions of the paper are ​highly relevant to the broader scientific literature. The integration of diffusion models with reinforcement learning for TSC tasks is a novel approach that addresses the limitations of existing methods, which often fail to handle both adversarial attacks and missing data simultaneously. The proposed framework builds on recent advancements in diffusion models and RL, extending their applications to the domain of traffic signal control. The paper also contributes to the growing body of research on ​robust RL​ and ​adversarial defense​ in real-world systems, providing a practical solution for improving the security and reliability of TSC systems.

**Theoretical Claims:**

The theoretical claims are ​solid and well-grounded. The paper builds on the foundations of diffusion models and reinforcement learning, providing a clear theoretical framework for the proposed methods. The authors discuss the ​forward and reverse processes​ of diffusion models and how they can be adapted for TSC tasks. The theoretical basis for the ​denoise​ and ​repaint​ algorithms is also well-explained, with references to existing literature on diffusion models and adversarial attacks. The theoretical claims are supported by the experimental results, demonstrating the practical applicability of the proposed methods.

---

> ### Author Rebuttal · Authors · 2025-03-30
>
> First and foremost, we sincerely thank you for pointing out the issues, as your suggestions are invaluable in enhancing the quality of this paper.
>
> 1. Scalability: The paper mentions that the computational cost increases with the number of intersections. Could the authors elaborate on potential strategies to scale RobustLight for large-scale urban networks with hundreds of intersections?
>
> Here, we need to clarify that our method can be deployed in either centralized or decentralized scenarios. In the case of centralized deployment, the model inference speed may slow down as the number of intersections increases.
>
> **Distributed Deployment**:  Specifically, if a distributed deployment is chosen, our diffusion model can be trained on a central server to fully leverage data from different intersections, while model parameter updates are performed on edge computing devices, similar to a federated learning architecture.
>
> **Centralized Deployment**: If a centralized deployment is preferred, scalability and real-time requirements for hundreds of intersections must be addressed. On the hardware side, we can procure high-performance servers or employ inference acceleration techniques, such as data partitioning and parallelized inference, to enhance the diffusion model's inference performance. On the algorithmic side, we can utilize techniques like DDIM [1] to accelerate the diffusion model's inference process.
>
> In summary, by carefully selecting the deployment strategy and leveraging advanced hardware and algorithmic optimizations, we can ensure the robustness, scalability, and real-time performance of our Robust TSC system while maintaining its security against potential attacks.
>
> [1] Denoising Diffusion Implicit Models.
>
> 2. Real-World Deployment: While the experiments are conducted on real-world datasets, the paper does not discuss the challenges of deploying RobustLight in real-world TSC systems. What are the practical considerations and potential barriers to real-world implementation?
>
>
> Our framework could leverage DDIM and high-performance GPUs for accelerated inference. Denoising supports decentralized deployment with centralized training, while data missing scenarios use centralized inference. DDIM acceleration times on one A100:
>
>
> | Deployment          | Scenario          | JN1   | HZ1   | NY1   |
> |---------------------|-------------------|-------|-------|-------|
> | Distributed       | U-rand noise| 0.098s| 0.097s| 0.096s|
> | Distributed       | Data missing| 0.129s| 0.115s| 0.131s|
> | Centralized         | U-rand noise| 0.364s| 0.431s| 2.43s |
> | Centralized         | Data missing| 0.123s| 0.164s| 0.81s |
>
>
> As shown, the total processing time is less than 1 second meet real-time control. However, it is important to note that in decentralized deployment scenarios, the repaint algorithm will not be able to address the **kriging missing (full-intersection failure)** issue within data missing. Distributed deployment significantly improves algorithm performance due to the reduction in the batch size of inference data.
>
> **We recommend deploying the centralized solution to address both kriging (full-intersection failure) and random missing (sensor-specific-direction-single-intersection-failure) data issues using data missing  algorithm, while employing data noise algorithms on the edge side using low-cost hardware to achieve accelerated denoising.**

---

### Official Review · Reviewer_ACd7 · 2025-03-12

**Overall Recommendation:** 3

**Summary:**

This paper point out the current challenge in the TSC systems, which include significant performance degration, limitation of existing defense methods and lack of online ability. To address these issues, authors propose RobustLight, a framework to enhance the robustness of online TSC systems, consisting of a TSC agent and a dynamic state filling (DSI) agent. This frame work contains 2 algorithms, denoise and repaint to defend against missing data and adversarial attack. Extensive experiments show the effectiveness of their framework.

**Claims And Evidence:**

The effectiveness of RobustLight is shown clearly in the experiment results in different datasets and different attacks.

**Essential References Not Discussed:**

NA

**Experimental Designs Or Analyses:**

For the experiment part, my concern is whether this method is still effective under some potential adaptive attacks, such as attacks effective on Diffusion Models.

**Methods And Evaluation Criteria:**

The benchmark used by their experiment session makes sense.

**Other Comments Or Suggestions:**

NA

**Other Strengths And Weaknesses:**

My another concern about this method, which is also stated in this paper, is the latency issue of this method because of the usage of diffusion models and whether it can handle traffic problems in a real-time way.

**Questions For Authors:**

My questions are 2 concerns mentioned in the previous sections.

**Relation To Broader Scientific Literature:**

I think the problem studied by this paper has a wide application in the real world traffic management.

**Theoretical Claims:**

Although this paper focuses on application side, the theoretical explanation is clear and the Lemma 3.1 and algorithm boxes are clear.

---

> ### Author Rebuttal · Authors · 2025-03-30
>
> First and foremost, we sincerely thank you for pointing out the issues, as your suggestions are invaluable in enhancing the quality of this paper.
>
> 1. For the experiment part, my concern is whether this method is still effective under some potential adaptive attacks, such as attacks effective on Diffusion Models.
>
> Diffusion Models are generally robust due to their iterative denoising process, which inherently introduces noise and reduces the impact of adversarial perturbations. However, like other deep learning models, they are not immune to adaptive attacks specifically designed to exploit their weaknesses.
>
> This situation resembles a Russian nesting doll, but based on our experience in TSC deployments, most TSC control systems are deployed within internal networks. While outdoor sensing devices are prone to noise interference and sensor damage, the TSC control module and our diffusion module can be deployed in either a distributed manner (with control devices at each intersection connected via an internal network) or a centralized manner (deployed within the internal network). This network isolation effectively protects the diffusion module from potential attacks.
>
> **Distributed Deployment**:  Specifically, if a distributed deployment is chosen, our diffusion model can be trained on a central server to fully leverage data from different intersections, while model parameter updates are performed on edge computing devices, similar to a federated learning architecture.
>
> **Centralized Deployment**: If a centralized deployment is preferred, scalability and real-time requirements for hundreds of intersections must be addressed. On the hardware side, we can procure high-performance servers or employ inference acceleration techniques, such as data partitioning and parallelized inference, to enhance the diffusion model's inference performance. On the algorithmic side, we can utilize techniques like DDIM [1] to accelerate the diffusion model's inference process.
>
> In summary, by carefully selecting the deployment strategy and leveraging advanced hardware and algorithmic optimizations, we can ensure the robustness, scalability, and real-time performance of Robust TSC system while maintaining its security against potential attacks.
>
> [1] Denoising Diffusion Implicit Models.
>
> 2.The latency issue of this method because of the usage of diffusion models and whether it can handle traffic problems in a real-time way.
>
>
> Our framework could leverage DDIM and high-performance GPUs for accelerated inference. Denoising supports decentralized deployment with centralized training, while data missing scenarios use centralized inference. DDIM acceleration times on one A100:
>
> | Deployment          | Scenario          | JN1   | HZ1   | NY1   |
> |---------------------|-------------------|-------|-------|-------|
> | Distributed       | U-rand noise| 0.098s| 0.097s| 0.096s|
> | Distributed       | Data missing| 0.129s| 0.115s| 0.131s|
> | Centralized         | U-rand noise| 0.364s| 0.431s| 2.43s |
> | Centralized         | Data missing| 0.123s| 0.164s| 0.81s |
>
> As shown, the total processing time is less than 1 second meet real-time control.  However, it is important to note that in decentralized deployment scenarios, the repaint algorithm will not be able to address the **kriging missing (full-intersection failure)** issue within data missing. Distributed deployment significantly improves algorithm performance due to the reduction in the batch size of inference data.
>
> **We recommend deploying the centralized solution to address both kriging (full-intersection failure) and random missing (sensor-specific-direction-single-intersection-failure) data issues using data missing  algorithm, while employing data noise algorithms on the edge side using low-cost hardware to achieve accelerated denoising.**

---

### Official Review · Reviewer_qV9c · 2025-03-13

**Overall Recommendation:** 4

**Summary:**

The paper introduces RobustLight, a novel framework designed to enhance the robustness of Traffic Signal Control (TSC) systems against adversarial attacks and missing data. The key contribution of RobustLight is the integration of an improved diffusion model into TSC, which enables real-time recovery of noisy or missing traffic data without altering the existing TSC algorithms. The framework consists of two main components: the Dynamic State Infilling (DSI) algorithm, which trains the diffusion model online, and two auxiliary algorithms, Denoise and Repaint, which leverage the trained diffusion model to address adversarial attacks and missing data, respectively.

## update after rebuttal
Rebuttal acknowledged.

**Claims And Evidence:**

The claims made in the submission are generally supported by clear and convincing evidence, particularly through extensive experimental results and visualizations. However, this paper mentioned online training and real-time implementation, but it does not provide sufficient theoretical nor experimental results to validate these claims. While the paper demonstrates the effectiveness of RobustLight in recovering noisy or missing data and improving traffic signal control performance, it does not explicitly show the complexity of the framework or how the framework performs in a real-time, online setting.

**Essential References Not Discussed:**

This paper lacks comparison with other recent methods. The latest method compared in the paper is proposed in 2022. Other TSC methods (e.g., GNN-based or Transformer-based approaches) or robust RL methods (e.g., adversarial training, self-supervised learning) can be compared and discussed (e.g., Explainable Deep Adversarial Reinforcement Learning Approach for Robust Autonomous Driving).

**Experimental Designs Or Analyses:**

The experiments are well-designed and provide convincing evidence to support the claims made in the paper.

**Methods And Evaluation Criteria:**

The proposed methods and evaluation criteria in the paper make sense for the problem of enhancing the robustness of Traffic Signal Control (TSC) systems against adversarial attacks and missing data.

**Other Comments Or Suggestions:**

- It is suggested that related works be added back to the main body of the paper to provide background knowledge; now, it appears in the Appendix.

**Other Strengths And Weaknesses:**

Strengths:
- This paper proposes a novel framework, "RobustLight." It introduces the Diffusion Model into the field of traffic signal control (TSC) to deal with data noise and missing value problems. Diffusion model has been widely used in the field of image generation, but it is an innovative attempt to apply it to TSC system to enhance robustness, especially in real-time online processing.
- RobustLight is not only able to handle a single type of attack (such as Gaussian noise), but also multiple complex attack types (such as MAD and MinQ attacks), and is able to deal with the problem of missing data caused by sensor corruption. This integrated defense capability is rare in current TSC systems, demonstrating the broad usage and robustness of the framework.
- The experiments are extensive. They are conducted on multiple real-world data sets to verify the effectiveness of RobustLight. Also, the visualization methods such as t-SNE visualization and violin diagram are used to demonstrate the effectiveness of state recovery, which enhances the reliability of experimental results.


Weaknesses:
- The code is not available.

-  Though the authors discuss the detection method,  the performance in a normal situation without an adversarial attack is not presented.  The authors discussed the detection of outliers in the paper. However, if the noise attack changes the queue length from 3 to 5, how can this be detected with outlier detection?

**Questions For Authors:**

1. Where is the correct code link?

**Relation To Broader Scientific Literature:**

The key contributions of the paper are related to and build upon several areas of the broader scientific literature, including Traffic Signal Control (TSC), Reinforcement Learning (RL), Robust RL, and Diffusion Models.

**Theoretical Claims:**

This paper focuses on experimental results and algorithmic contributions rather than theoretical proofs. This paper provides a detailed description of the proposed methods, including the Dynamic State Infilling (DSI) algorithm, Denoise algorithm, and Repaint algorithm. This paper does not include formal theoretical claims or proofs. Instead, the paper relies on experimental results to demonstrate the effectiveness of the proposed RobustLight framework.

---

> ### Author Rebuttal · Authors · 2025-03-30
>
> First and foremost, we sincerely thank you for pointing out the issues, as your suggestions are invaluable in enhancing the quality of this paper.
>
> W1:  https://anonymous.4open.science/r/RobustLight-72B2/README.md
>
> W2：Traffic movements and TSP are defined in Figure 1 and Section 2.1, with cyan coloring used for aesthetic design.  D() represents the norm distance.
>
> W3&W6:
>
> We conduct the experiments for Difflight[1] and MissingLight[2]. [1] uses offline RL and diffusion models for two missing-data cases: **random missing** (single-intersection-sensor failure) and **kriging missing** (full-intersection failure), requiring separate treatments. To validate our approach, we setup involved randomly masking data from Kriging Missing (12.5%) and Random Missing (12.5%).
>
>
> ### Data Missing , Our is Based On Advanced-Colight
>
> | Method                     | JN1             | HZ1             |
> |----------------------------|-----------------|-----------------|
> | [2]     | 354.73  | 348.68  |
> | [1]                 | 353.45     | 346.05  |
> | **Our** | **320.31** | **296.69** |
>
>
> ### Data Noise Scale 3.5
>
> | Method                     | JN1             | HZ1             |
> |----------------------------|-----------------|-----------------|
> | [1]-MinQ           | 321.98  | 426.14  |
> | **Our** | **283.13** | **397.32** |
> | [1]-MAD            | 384.71  | 366.14  |
> | **Our** | **323.25** | **309.24** |
>
> As can be seen, our method not only effectively handles data missing scenarios in both (**kriging missing and random missing**) but also performs well in data noisy condition.
>
> ### W6：Without Data Missing and Data Noise
> | Dataset | DiffLight | Advanced-Colight |
> |---------|-----------|------------------|
> | JN1     | 268.43    | **245.73**       |
> | HZ1     | 283.92    | **270.45**       |
>
>
> Under clean data, RobustLight preserves the optimal performance of the TSC algorithm without unnecessary denoising, it simplifies integration and enhancing system stability and reliability.
>
> Compared to [3], which uses adversarial pre-training to improve robustness of RL, but it cannot repair corrupted inputs and risks overfitting to specific attacks, our diffusion model handles diverse disturbances without attack-specific training, directly denoising states for robust RL inputs as a plug-and-play module.
>
>
> [1] DiffLight: A Partial Rewards Conditioned Diffusion Model for Traffic Signal Control with Missing Data. NIPS 2025
>
> [2] Reinforcement Learning Approaches for Traffic Signal Control under Missing Data. IJCAI 2023
>
> [3] Explainable Deep Adversarial Reinforcement Learning Approach for Robust Autonomous Driving. TIV 2024
>
> W4:
>
> Our framework could leverage DDIM and high-performance GPUs for accelerated inference. Denoising supports decentralized deployment with centralized training, while data missing scenarios use centralized inference. DDIM acceleration times on one A100:
>
> | Deployment          | Scenario          | JN1   |  NY1   |
> |---------------------|-------------------|-------|-------|
> | Decentralized       | U-rand noise| 0.098s|  0.096s|
> | Centralized         | Data missing| 0.123s|  0.81s |
>
> As shown, the total processing time is less than 1 second meet real-time control. Further deployment discussion can be found in reviewer Jqz812.
>
> W5:
>
> Our RobustLight addresses challenges in TSC deployment, where TSC RL inputs often suffer from noise or missing values, impacting control. To handle the dynamic nature of traffic flow, we propose an online training framework using diffusion models as the upper level RL policy during clean data periods. When noise or missing data is detected, our trained DSI agent employs denoising and repainting algorithms to restore data quality, improving lower level TSC RL performance.
>
> W6&W7：
>
>  [1] introduced TP-FDS, which identifies anomalies by comparing new data distributions with historical data from the same period, achieving an AUC of 96% and an F1 score of 76%. Minor changes like 3-5 have minimal impact on system efficiency. Anomalies can also be detected by cross-referencing data from multiple sensors, such as cameras and radar. Rule-based methods are another option; for example, a queue increase from 3 to 5 during a north-south green light would signal an anomaly. To simulate real-world scenarios, we conducted experiments with detection rates of 80% and 60%:
>
> ### MAD Scale 3.5
> | Detection Rate | Base   | ATT | Base| Throughput | Dataset |
> |-----------------|-----|-----|-----|------------|----------|
> | 80%             | 487| **297**      | 5812  |**6154**   | JN1      |
> |            |  463   | **326**   |2888 | **2938**         | HZ1      |
> | 60%            | 487|  **328**   | 5812|**6131**         | JN1      |
> |             |     463   |**331**  |2888| **2930**          | HZ1      |
>
> It sustains high performance with different detection rates.
>
> Refer to Jqz8 and ACd7, our theoretical explanation based Lemma 3.1
>
> [1] Traffic Anomaly Detection: Exploiting Temporal Positioning of Flow-Density Samples

---

### Official Review · Reviewer_YPk8 · 2025-03-13

**Overall Recommendation:** 3

**Summary:**

This paper focuses on a very interesting problem. For the data missing problem faced in traffic signal control, the authors use the diffusion model to complete and clean the data. The experimental results show that this method effectively improves the control performance of the reinforcement learning model in data missing or contaminated environments.

**Claims And Evidence:**

i) Experimental results on some datasets are not reported.
ii) Lack of baseline comparison for data generation methods.

**Essential References Not Discussed:**

The authors put the related work section in the appendix and did not discuss similar data generation methods.

**Experimental Designs Or Analyses:**

The comparison method is not perfect and some experimental results are not reported.

**Methods And Evaluation Criteria:**

Traffic uses common standards in the field. There is a lack of quantitative standards for data denoising and completion.

**Other Comments Or Suggestions:**

i) Other related data generation methods are introduced to reflect the innovation of the proposed data generation method.
ii) Give complete experimental results
iii) Update the link to the code repository
iv) Provides quantitative analysis results of data generation methods and comparative experimental results of generation quality

**Other Strengths And Weaknesses:**

Strengths:
i) The authors have addressed a new issue
ii) Use the new diffusion model to complete missing data and denoise

Weaknesses:
i) The authors do not fully introduce other related data generation methods
ii) The experimental part lacks experimental results on some datasets
iii) There is no quantitative analysis of the quality of the generated data, nor is there any comparison with baseline methods.

**Questions For Authors:**

i) What is the difference between the proposed diffusion model-based data generation method and the existing diffusion models?

**Relation To Broader Scientific Literature:**

Provides a new perspective for studying traffic signal control issues.

**Theoretical Claims:**

This paper does not provide any theoretical proof.

---

> ### Author Rebuttal · Authors · 2025-03-28
>
> First and foremost, we sincerely thank you for pointing out the issues. Your suggestions are invaluable in enhancing the quality of this paper. Below is our answer to your questions.
>
> 1. Experimental results on some datasets are not reported.
>
> Due to space constraints, data noise results for JN2 and HZ2 were omitted from the main text. The partial snapshot below shows RobustLight’s superior performance. Additional dataset results are included in the appendix and link.
>
> ### Advanced-CoLight  Noise Scale 3.5
> | Dataset    | Noise Type | base| RobustLight |
> |------------|------------|--------------------------|-------------------------------|
> |     JN1       | MAD        | 325.87                   | **271.58**                    |
> |            | MinQ       | 323.07                   | **300.31**                    |
> |     JN2       | MAD        | 405.37                   | **359.03**                    |
> |            | MinQ       | 399.87                   | **372.84**                    |
> |      NY1      | Mask25\%| 1246.85                   | **1086.77**                    |
> |      NY2      | Mask25\%| 1430.49                   | **1290.79**                    |
>
> 2. Other related data generation methods are introduced to reflect the innovation of the proposed data generation method.
>
> HINT[1] proposed using Transformer combined with GAN for data reconstruction, while CaPaint[2] employs diffusion for data generation. However, these methods require extensive offline data for pre-training and only address data missing, not data noise. DiffLight[3] tackles random missing and kriging missing but uses separate algorithms for each and underperforms SOTA on clean data. In contrast, our method employs a single model to simultaneously resolve data missing and noise, handling both random missing and kriging missing with a unified algorithm. Moreover, our approach acts as a plug-in, optimizing SOTA algorithms without altering their core structure.
>
> [1] HINT: High-quality INpainting Transformer with Mask-Aware Encoding and Enhanced Attention
>
> [2] Causal Deciphering and Inpainting in Spatio-Temporal Dynamics via Diffusion Model
>
> [3] DiffLight: A Partial Rewards Conditioned Diffusion Model for Traffic Signal Control with Missing Data
>
> 3. Missing baseline and quantitative analysis.
>
> We fine-tuned the HINT model on an offline dataset and evaluated it on the TSC task, quantifying performance using PSNR (higher is better) and MAE (lower is better). Our algorithm surpasses HINT in data generation capability.
>
> ### Data Generation Performance Comparison
> | Dataset | Method      | PSNR      | MAE      | ATT      |
> |---------|-------------|-----------|----------|----------|
> | JN1     | HINT        | 9.78      | 1.11     | 396.24   |
> |         | **RobustLight** | **10.46** | **0.66** | **298.35** |
> | JN2     | HINT        | 18.88     | 1.36     | 283.98   |
> |         | **RobustLight** | **24.07** | **1.20** | **259.56** |
> | JN3     | HINT        | 10.24     | 1.33     | 395.62   |
> |         | **RobustLight** | **11.71** | **1.15** | **301.63** |
> | HZ1     | HINT        | 11.90     | 0.88     | 371.74   |
> |         | **RobustLight** | **13.90** | **1.01** | **328.26** |
> | HZ2     | HINT        | 5.35      | 0.76     | 384.67   |
> |         | **RobustLight** | **6.53**  | **0.85** | **375.64** |
> | NY1     | HINT        | 16.31     | 1.67     | 1189.56  |
> |         | **RobustLight** | **17.25** | **1.51** | **1086.77** |
> | NY2     | HINT        | 13.39     | 1.53     | 1394.96  |
> |         | **RobustLight** | **15.65** | **1.29** | **1290.79** |
>
> To validate our approach, we setup involved randomly masking data from **Kriging Missing (12.5\%)** and **Random Missing (12.5\%)**.  AMPR (Advanced-MaxPressure-Robustlight), ACR (Advanced-Colight-Robustlight).
>
> ###  Performance Compared with DiffLight
> | Dataset | Method                  | Noise/Mask | PSNR      | MAE      | ATT      |
> |---------|-------------------------|------------|-----------|----------|----------|
> | JN1     | DiffLight               | U-rand     | 6.71      | 6.26     | 310.92   |
> |     | **AMPR**  |    | **7.20**      | **5.42**     | **304.34**   |
> | HZ1     | DiffLight               |     | 6.85      | 7.93     | 361.31   |
> |     |**AMPR** |     | **7.71**      | **5.12**     | **297.34**   |
> | JN1     | DiffLight               | 25%        | 7.66      | 1.05     | 366.05   |
> |    | **ACR** |      | **9.34** | **0.89** | **304.13** |
> | HZ1     | DiffLight               |     | 18.05      | 1.84     | 372.53   |
> |    | **ACR** |    | **22.96** | **1.17** | **306.56** |
>
> 4. The difference with other diffusion model.
>
> Key modifications: (1)use a new beta schedule (2) optimized with new loss function. (3) employ new action gradient method to improve data noise and missing.
>
> 5. New link: https://anonymous.4open.science/r/RobustLight-72B2/README.md
>
> 6. More baseline results refer to qV9c and code link.
>
> 7. Refer to Jqz8 and ACd7,  our theoretical explanation based Lemma 3.1

---

### Decision · Program_Chairs · 2025-05-01

**Decision:**

Accept (poster)

**Comment:**

This paper introduces a novel framework for utilizing a diffusion model to enhance the robustness of traffic signal control systems against adversarial attack and missing data.

Among the paper's strengths are the following: (1) the framework makes novel use of diffusion models to address an important practical problem; (2) the paper's claims are supported by an extensive experimental study utilizing multiple real-world data sets which gives clear and convincing evidence; and (3) the approach is capable of handling multiple complex attack types.

With respect to weaknesses: (1) there is some concern about the latency of the method and whether it can handle the real-time aspects of the traffic signal control domain, and (2) the significance of the results are weakened somewhat by the fact that diffusion models are not immune to adaptive attacks specifically designed to exploit their weaknesses, and (3) the paper lacks comparison with some more recent advances in diffusion models and reinforcement learning.

Overall, the paper makes a nice contribution and the consensus recommendation is that the paper be accepted. Please be sure to address all comments and suggestions made by the reviewers when preparing the final version of the paper.